# A meta-analytic review and conceptual model of the antecedents and outcomes of goal adjustment in response to striving difficulties

There is growing interest in how and why individuals adjust their goals in response to difficulties encountered during goal striving and the outcomes of such adjustments; however, research on these topics is fragmented across theoretical perspectives and life domains. To address this issue, we conducted a systematic search of databases (Web of Science, Scopus, PsycInfo, Business Source Ultimate, Proquest Dissertations and Theses Global, Medline; last updated May 2025) of empirical studies examining antecedents or outcomes of goal adjustment. Studies were eligible if they examined predictors or wellbeing/functional/goal-related outcomes of goal disengagement, goal reengagement, or goal-striving flexibility. We identified 1,421 effect sizes from 235 studies, which we categorized and mapped onto a conceptual model. Further, we used random-effects meta-analyses to examine the strength and direction of associations between model categories and goal adjustment variables. Despite relatively high-quality ratings (assessed using QualSyst), the overall standard of accumulated evidence was determined to be low to moderate due to the reliance on cross-sectional studies, risk of publication bias and high heterogeneity. Nonetheless, we identified associations between multiple antecedent categories and goal disengagement, reengagement and flexibility, as well as associations between these different aspects of goal adjustment and wellbeing, functional and goal-related outcomes. We conclude that different aspects of goal adjustment are predicted by unique combinations of antecedent variables, and predict distinct outcomes. Our conceptual model consolidates the literature on goal adjustment and provides a roadmap for a more systematic investigation of this field going forward.

In the pursuit of personal and professional aspirations, individuals frequently encounter obstacles, setbacks, or shifting circumstances that challenge the attainability or desirability of their goals. In response to these difficulties, people may persist in their efforts, or they may adjust their goals by modifying, reprioritizing or abandoning them altogether. Although the benefits of persistence have long been emphasized[1], a growing body of research has highlighted the adaptive value of adjustment when goals become unattainable or overly costly to pursue[2,3].

Despite this empirical growth, the literature on goal adjustment remains fragmented. Researchers have approached the topic from a variety of theoretical perspectives and have examined a wide range of potential predictors and outcomes of goal adjustment across diverse life domains. Yet, so far, meta-analytic efforts have primarily focused on associations between various goal adjustment capacities and wellbeing[4,5]. This work, although important, has not considered other critical issues such as antecedents of goal adjustment, and the

e-mail: n.ntoumanis@bham.ac.uk

effects of goal adjustment on goal striving and day-to-day physical or social functioning.

In this Article, we address these knowledge gaps by presenting a meta-analytic synthesis and integrative conceptual model of goal adjustment, its antecedents and its outcomes. Specifically, we (1) organize and synthesize empirical findings on the antecedents and outcomes of goal adjustment across diverse domains; (2) broaden the scope of outcomes examined to include not only psychological wellbeing but also goal striving, physical health and social functioning; and (3) use the resulting evidence base to propose a model of goal adjustment that integrates findings across multiple theoretical perspectives. In doing so, we provide a synthesis that advances understanding of the factors that influence goal adjustment and its multifaceted outcomes.

## What is goal adjustment?

Scobbie et al.[6] attempted to map definitions and theoretical underpinnings of goal adjustment processes. The authors identified three relevant perspectives: the dual process model[7], the goal adjustment model[3,8] and control-theory-informed models[9–11]. In addition, they identified three subcategories into which goal adjustment processes can be grouped: goal disengagement (letting a goal go), goal reengagement (setting a new goal) and goal adjustment (various processes that enable modification of existing goals to render them more achievable). We note that Scobbie et al. used 'goal adjustment' both as a subcategory and as an umbrella term to refer to all three subcategories.

An alternative approach[12] aggregated the dual process model[13], the model of selection, optimization and compensation[14], and the motivational theory of lifespan development[15] to highlight three comparable constructs: goal engagement, goal disengagement and metaregulation (flexibly optimizing goals to match changing circumstances and opportunities). This work underscores the dynamic interplay between persistence and flexibility, and reflects a broader attempt to unify perspectives on how individuals regulate their goals over the life course. On the other hand, ref. 6 offers a structured taxonomy for organizing research on adjustment[12] and provides a theoretically integrated account of core self-regulatory mechanisms.

Building on these research efforts, we propose that three core adjustment processes capture how individuals alter their goals and their general propensity towards adjusting goals. We use 'goal adjustment' to refer to the overarching concept that encompasses the three processes: 'goal disengagement', 'goal reengagement' and 'goal-striving flexibility'. We adhere to the definitions of goal disengagement and reengagement presented in the previous paragraphs, and define goal-striving flexibility as an individual's general disposition to remain open to modifying their striving, cognitions or goals to better align with their capabilities, resources and situational demands. This distinction has implications for how these constructs have been studied in the literature. Disengagement and reengagement are often assessed with respect to specific goals at the behavioural or situational level. That is, authors are often concerned with how individuals react to unattainability in a certain domain or situation. Conversely, examinations of flexibility have been more interested in how people adapt to setbacks generally or over a protracted timescale. Disengagement, reengagement and flexibility have been broadly captured by various theoretical perspectives that have been put forth to explain goal adjustment. However, by focusing on disengagement, reengagement and flexibility as discrete processes, these accounts have somewhat overlooked the potential interplay between dispositional characteristics and situational goal adjustment. We argue that an integrated perspective is needed to clarify the similarities and differences among these goal adjustment processes.

## Theoretical conceptualizations of goal adjustment

In their early control-theory-informed account of goal adjustment processes, ref. 9 posited that "an impulse to withdraw or disengage from the [goal] may occur if the person's expectancy of being able to reduce the discrepancy [between their current state and behavioural ideal] is sufficiently unfavourable" (p. 121). Contemporaneously, ref. 16 drew a distinction between primary control processes, which aim to change the environment to align it with the individual's wishes, and secondary control processes in which the individual changes themselves to align with environmental imperatives. These early accounts emphasized the idea that goal adjustment occurs when there is an irreconcilable mismatch between an individual's current and desired state, and that in such cases, abandoning or changing goals is necessary for ongoing functioning.

Informed by research on how people manage age-related setbacks, the dual process model of ref. 7 introduced the concept of accommodative coping, formalizing the adaptive capacity of goal adjustment. Accommodative coping[13] refers to a coping approach in which the individual overcomes obstacles over the course of development by adjusting personal preferences and goal orientations to situations. Examples of accommodative coping include adjustments of one's aspiration level, revision of value priorities and evaluative standards, and cognitive reappraisals of the situation. These processes have been proposed[13] to produce radical and ultimately beneficial changes in an individual's beliefs about themselves and the world, thereby facilitating healthy development over the lifespan. Extending this line of reasoning, refs. 12,15 both highlighted the cycle of goal selection, pursuit and relinquishment that repeats across a person's lifetime as control capacities and circumstances fluctuate, arguing that a flexible approach to goal striving allows individuals to maintain day-to-day functioning in the face of developmental changes. Our concept of 'flexible goal striving' broadly captures this idea, namely, that each person maintains, to a greater or lesser degree, a general disposition towards adjusting their goals to accommodate changing circumstances.

The turn of the twenty-first century witnessed a discourse around mechanisms that underpin the adjustment of specific goals. Wrosch and colleagues[3,8] emphasized disengagement and reengagement as two processes integral to the active modification of individual goals. As mentioned previously, goal disengagement refers to the intentional process of letting go of unattainable goals. It involves accepting reality, acknowledging one's limitations and ceasing efforts to pursue the goal. When individuals encounter persistent obstacles, setbacks or changing circumstances that make their current goals unrealistic or unachievable, they probably experience negative emotions such as frustration, disappointment or sadness[17]. Disengaging can protect individuals by sheltering them from further distress[18].

Letting go of unattainable goals also enables people to redirect their efforts towards alternative goals that are more feasible, desirable and aligned with their current circumstances and capabilities. Reengagement with a new or modified goal is the adaptive process that allows one to restore a sense of purpose and motivation, promoting positive emotions and wellbeing[19–21]. Reengagement can occur in various ways. An individual can choose an alternate means to achieve the same higher-order goal, form a new higher-order goal, or scale back to a reduced version of the original goal. All three options retain the potential to produce positive outcomes[11].

Although some strands of the goal adjustment literature conceptualize disengagement and reengagement as trait-like tendencies[3], other research has emphasized the situational and goal-specific nature of these processes[22,23]. This distinction between dispositional tendencies and context-dependent responses is critical for advancing theoretical and empirical understanding. Trait-level measures capture enduring individual differences in how people typically respond to goal-related challenges, offering valuable insights into broad patterns of self-regulation. However, such measures may overlook the dynamic, context-sensitive adjustments that individuals make in response to specific goal failures or life events, patterns more effectively captured by goal-specific assessments. A more complete

understanding of goal adjustment therefore requires attention to both the predictors of stable tendencies and the situational factors that shape goal-specific behaviour.

## Outcomes of goal adjustment

A commonality among goal adjustment theories is the notion that adjustment results in benefits for personal wellbeing and functioning. Empirical evidence appears to support this proposition. A meta-analysis ($N = 28,145$)[5] of the association between accommodative coping and psychological wellbeing showed a large—sensu ref. 24—effect size ($r = 0.32$). Similarly, another meta-analysis ($N = 7,241$)[4] reported small-to-moderate associations between wellbeing and disengagement ($r = 0.08$), as well as reengagement ($r = 0.19$). Theoretical perspectives have also highlighted the relevance of goal adjustment for both physical and societal functioning. Across the lifespan, the ability to flexibly modify personal goals has been identified as a critical adaptive resource. For instance, goal adjustment plays a central role in coping with age-related declines in physical capacity[13,15] and the onset of chronic health conditions[25], and in the successful management of social relationships in adulthood[26]. Similarly, adaptive goal adjustment has been linked to effective preparation for retirement[27] and to navigating career transformations[28]. The capacity to recalibrate or disengage from obstructed goals during major life transitions or unexpected events enables individuals to maintain functional autonomy and psychological wellbeing in a rapidly changing world. Conversely, failure to adapt one's goals in the face of shifting circumstances can compromise one's ability to manage emergent demands across the life course.

Much of the primary research and evidence synthesis in the goal adjustment literature has focused on relations between goal adjustment and wellbeing or functioning; however, goal adjustment also has implications for goal pursuit. Goal adjustment implies shifts in goal-directed behaviour and cognition, which are associated with changes in outcomes related to goal striving, such as goal commitment, pursuit or attainment. Evidence supports associations between cognitive goal disengagement and the reduction or cessation of goal-directed effort, as well as goal attainment[29]. Conversely, cognitive reengagement has been linked with the resumption of goal-directed behaviour and renewed goal progress[23], particularly if goal unattainability is realized in the early stages of striving[22].

Goal adjustment, goal-related outcomes and wellbeing are interconnected. Both refs. 3,8, along with ref. 11, emphasized that cognitively disengaging but behaviourally continuing goal pursuit, or behaviourally disengaging but maintaining cognitive attachment to the goal, will negatively influence wellbeing. Similarly, reengagement is most beneficial for wellbeing when the individual can experience progress and success with their reengaged goal[30]. Although previous meta-analyses have synthesized the associations between goal adjustment processes and psychological wellbeing[4,6], a systematic synthesis of the evidence concerning the goal-related and functional outcomes of goal adjustment is lacking.

## Antecedents of goal adjustment

Various attempts have been made to elucidate variables that prompt an individual to shift from a state of active goal pursuit to goal adjustment[31]. In their initial formulation, refs. 9,10 suggested that the transition to phases of goal adjustment is triggered when an individual identifies an insurmountable difference between their current and desired state. The action crisis model, developed by refs. 32–34, posits that if an individual fails to combat obstacles that accumulate during goal striving, they will feel dejected and disenfranchised with their goal. This disenfranchisement, in turn, conduces to weighing the costs and benefits of continued goal pursuit, and potentially adjusting the goal[30,35].

Other researchers have proposed a range of alternative variables that might also predict goal adjustment. For example, work informed by self-determination theory[36] has demonstrated that people find it more difficult to disengage from autonomously motivated goals, which are more valued because they align with one's interests or self-concept, compared with goals driven by controlled motives, which misalign with one's core self[22,37]. In a narrative review, ref. 2 criticized the lack of a comprehensive model of variables that contribute to goal adjustment as limiting the development of research on the field.

## The current study

Efforts to consolidate evidence on goal adjustment have comprised single life domains, used differing methods of quantifying adjustment, been grounded in divergent theoretical perspectives, or focused on distinct outcomes[4–6]. Although informative, such approaches are limited, as goal adjustment should theoretically benefit all life domains and entail cognitive, emotional and behavioural components that cannot be fully captured by a single theoretical perspective or measurement technique[38]. To gain a comprehensive understanding of the processes underpinning goal adjustment, a theory- and measurement-agnostic approach is required. Consequently, we collated research from any life domain that assessed the observable modification of goal striving (that is, behavioural adjustment) or featured any scale of self-reported goal adjustment capacity (that is, cognitive adjustment). We integrated research on goal-related, wellbeing and functional outcomes of goal adjustment, and provided a synthesis of its antecedents.

Our overall aim was to bring together the literature on antecedents and outcomes of various suggested goal adjustment capacities. We had three objectives. The first was to systematically search for variables identified as antecedents or outcomes of key goal adjustment capacities and consolidate these variables into overarching categories, thus establishing a foundation for a conceptual model of goal adjustment. The second objective was to assess the size and direction of associations between our proposed antecedent categories and goal adjustment capacities via meta-analysis. The third objective was to assess the size and direction of associations between goal adjustment capacities on our proposed outcome categories. Taken together, this work offers a unified picture of how various personal and environmental variables influence different goal adjustment capabilities, and how these capacities in turn contribute to goal striving and livelihood.

## Literature search overview

We depict the selection of studies for the meta-analyses as a PRISMA flow diagram in Fig. 1. In our initial search, we identified 188 articles that were eligible for inclusion. In our first updated search, we identified an extra 22 articles. In our second updated search, we identified an additional 25 studies. We extracted a total of 1,421 effect sizes from 281 samples. Studies focused primarily on disengagement or reengagement (goal disengagement, 26.3% of studies; goal reengagement, 13.9% of studies; combined disengagement and reengagement, 27.8% of studies), and secondarily on the general approach to being flexible in one's goal striving (29.9% of studies). A small percentage (2.5%) of studies tested both disengagement/reengagement and goal-striving flexibility. Effects related to goal disengagement (64%) and reengagement (77%) were predominantly measured with the respective subscales of the Goal Adjustment Scale[3], while effects related to goal-striving flexibility (78%) were predominantly measured using the flexible goal adjustment subscale of the Tenacious Goal Pursuit/Flexible Goal Adjustment Scale[13].

Of the effect sizes, 306 pertained to antecedents of goal disengagement, 186 to antecedents of goal reengagement, and 96 to antecedents of goal-striving flexibility. In terms of outcomes linked to goal disengagement, we located 72 effect sizes relating to goal progression/reduction outcomes, 33 for functioning/impairment outcomes, and 200 for wellbeing/illbeing outcomes. For goal reengagement, we located 56 effect sizes for goal progression/reduction outcomes, 31 for functioning/impairment outcomes, and 177 for wellbeing/illbeing outcomes. Finally, for flexible goal adjustment, we identified 30 effect sizes for

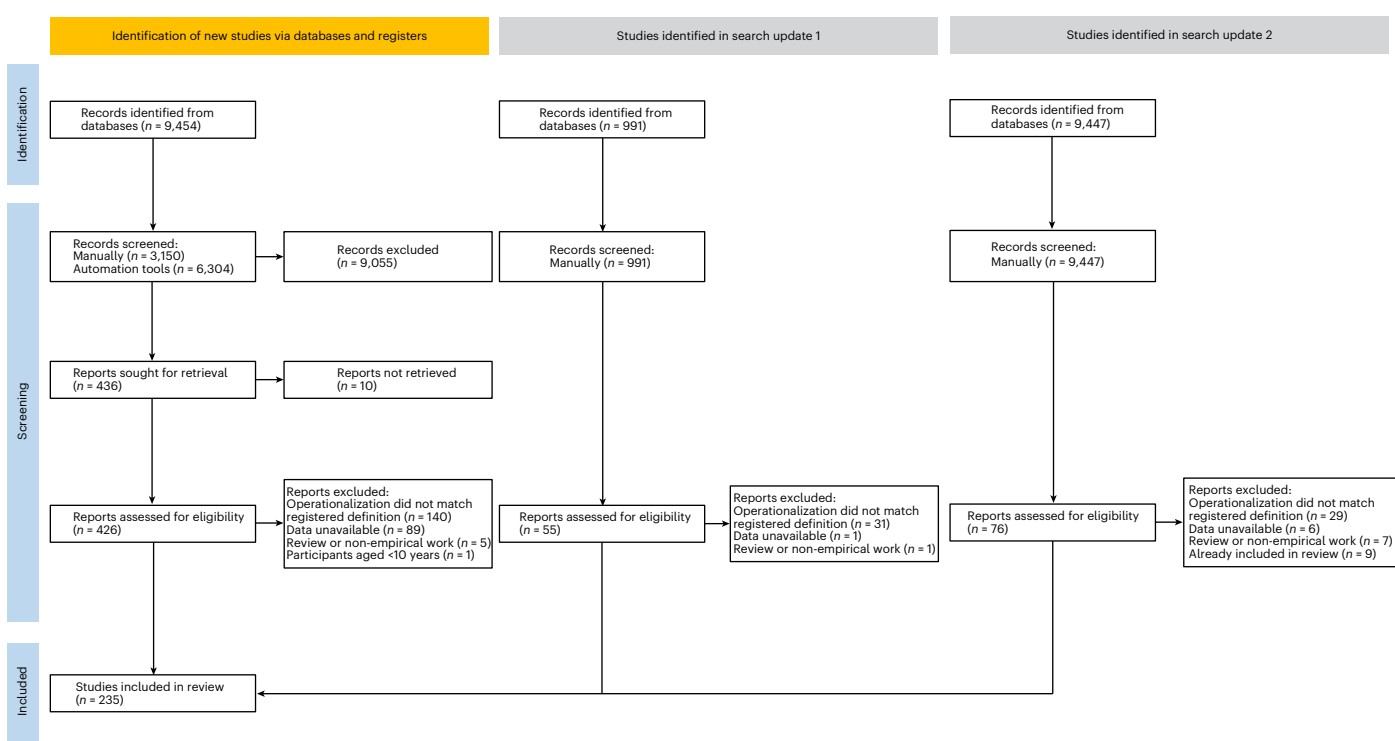

**Fig. 1 | PRISMA flow diagram.**

goal progression/reduction-related outcomes, 46 for functioning/impairment outcomes, and 188 for wellbeing/illbeing outcomes. We provide a full summary of included studies in Supplementary Table 5 and citations in Supplementary Information 3 (citations in the reference list marked with * were also included in the meta-analysis).

## Conceptual model

We depict our conceptual model in Fig. 2. Our model describes the antecedent/outcome categories as well as the direction of association between these categories and the three goal adjustment processes, as determined by the meta-analyses. In this section, we provide a narrative description of our model categories.

Agency antecedents shape an individual's level of control or influence over their goal striving. Agentic antecedent variables include both access to physical resources required to pursue goals (for example, time, money) and beliefs about one's ability to exert control over their goal striving (for example, control beliefs). Agency-inhibiting variables reduce agentic control over goal striving by making individuals feel powerless to seek alternatives or doubt that their actions can impact goal outcomes, whereas agency-enhancing variables provide avenues for exploration and experimentation during goal striving.

Goal-related antecedents define how individuals think and behave in relation to their goal, including their sources of motivation and the strategies they use to overcome obstacles during goal pursuit. Importantly, such antecedents encompass cognitions and behaviours directly tied to an individual's goals. Goal-enhancing antecedent variables are beneficial to goal striving (for example, autonomous motivation), whereas goal-inhibiting antecedent variables diminish a goal's desirability or attainability (for example, action crisis). Affective antecedents reflect an individual's emotions and mood. Aligning with the broader emotion literature, these can be either pleasant (for example, positive affect) or unpleasant (for example, negative affect) experiences or states. We classified pleasant affective variables as affect enhancing, and unpleasant variables as affect diminishing.

Context/developmental antecedents pertain to the wider environment in which individuals pursue their goals. These antecedents can reflect either the immediate context (that is, circumstances) surrounding goal pursuit or the historical context (that is, earlier life experiences) that shape an individual's approach to their goals. Positive context/developmental variables are seen as promoting healthy development and/or functioning (for example, autonomy-supportive environments) or inhibiting healthy development/goal striving (for example, age-related function loss). Finally, personality antecedents capture traits that are either adaptive or neutral for a person's functioning (for example, grit as an adaptive variable, openness to experience as a neutral variable). They also capture maladaptive facets of one's personality (for example, disorder).

The outcome variables we used are largely self-explanatory. Goal progression and goal reduction refer to variables capturing the effective versus ineffective pursuit or maintenance of personal goals (for example, goal progress vs self-handicapping). Wellbeing and illbeing encompass psychological indicators of mental health and emotional welfare, such as life satisfaction and depressive symptoms, respectively. In contrast, functioning and impairment reflect biological and social dimensions of health, including variables that describe individuals' physical capabilities and social integration (for example, experiencing burnout and forming supportive social networks). We distinguished mental wellbeing/illbeing from physical and social functioning/impairment on the basis of theoretical perspectives that posit unique pathways linking goal adjustment to different domains of functioning. For example, both the dual process model and lifespan development theory suggest that flexible goal adjustment plays a central role in adapting to age-related declines in physical capacity[13,39].

## Meta-analytic evidence for associations between model constructs

### Antecedent variables

We present meta-regression coefficients, confidence intervals and heterogeneity estimates ($I^2$) for all antecedent categories in Table 1. We present moderator analyses in Supplementary Information 1. Heterogeneity was high for all antecedents of goal disengagement and goal reengagement, as well as for most antecedents of goal-striving flexibility.

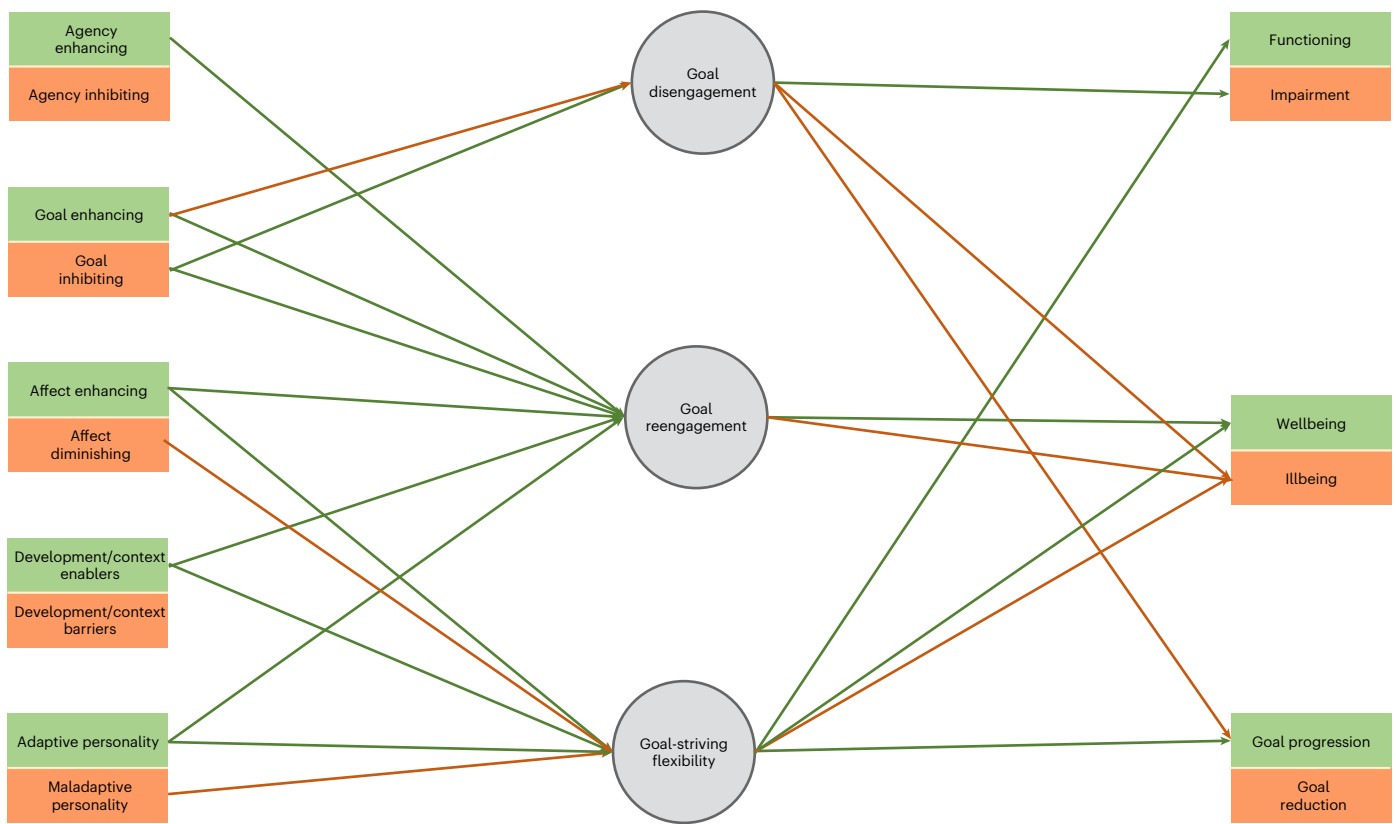

**Fig. 2 | Integrative model of goal adjustment.** Green lines represent significant (two-sided meta-regression, $p < 0.05$) positive associations, red lines represent significant (two-sided meta-regression, $p < 0.05$) negative associations. Exact $p$ values and confidence intervals are available in Tables 1 and 2.

**Antecedents of goal disengagement.** Goal-inhibiting variables positively predicted goal disengagement, with the effect size being small to moderate. We broke this category down into its contingent variables. Previous negative feedback regarding goal suitability ($r = 0.45$, 95% CI = [0.06, 0.91], $n = 837$, $k = 3$), threat appraisals ($r = 0.42$, 95% CI = [0.07, 0.83], $n = 1,342$, $k = 3$) and action crisis ($r = 0.36$, 95% CI = [0.21, 0.55], $n = 3,128$, $k = 7$) were significantly positively related to disengagement, whereas generalized negative feedback was negatively related to disengagement ($r = −0.66$, 95% CI = [−1.46, −0.12], $n = 597$ $k = 2$). We note, however, that some of these associations should be interpreted with caution, as we synthesized them from few (that is, $k \leq 3$) studies. Conversely, goal-facilitating variables were negatively related to goal disengagement, also with a small to moderate negative effect size. No individual variable from this category was significantly associated with goal disengagement. Sensitivity analysis including longitudinal and experimental studies only yielded a small to moderate positive association between goal-inhibiting variables and goal disengagement ($r = 0.16$, 95% CI = [0.06, 0.27], $n = 4,984$, $k = 37$), but not between any other antecedent variables.

**Antecedents of goal reengagement.** Affect-enhancing variables were positively related to goal reengagement, with the effect size being large. Examining individual affect-enhancing variables revealed that both optimism ($r = 0.41$, 95% CI = [0.13, 0.73], $n = 1,584$, $k = 9$) and positive affect ($r = 0.37$, 95% CI = [0.02, 0.73], $n = 794$, $k = 4$) were positively related to reengagement.

The categories adaptive personality traits and developmental/contextual enablers were also positively related to goal reengagement, with the effect size being moderate to large. When breaking adaptive personality traits into individual variables, no variable was significant, probably due to the small number of studies examining individual

variables ($k_{range} = 2–3$). We were unable to meta-analyse the individual developmental/contextual enabler variables due to a lack of variables within the category that had been involved in more than one study.

In addition, agency-enhancing and goal-enhancing variable categories were positively related to goal reengagement, with the effect size being moderate. No individual agency-enhancing variables were significantly related to goal reengagement, probably due to the limited number of studies and the diversity of independent variables used. Autonomous motives ($r = 0.22$, 95% CI = [0.16, 0.27], $n = 2,245$, $k = 14$), a positive orientation toward goal adjustment ($r = 0.26$, 95% CI = [0.10, 0.43], $n = 233$, $k = 3$) and completing mental contrasting with or without implementation intentions ($r = 0.30$, 95% CI = [0.04, 0.58], $n = 1,440$, $k = 6$)—all individual goal-enhancing variables—were significantly associated with goal reengagement. Finally, the goal-inhibiting variables category was negatively related to reengagement, with a small effect size. No individual goal-inhibiting variables were significantly linked to goal reengagement.

Results were largely unchanged in sensitivity analyses considering only longitudinal and experimental studies. Adaptive personality traits ($r = 0.27$, 95% CI = [0.04, 0.49], $n = 815$, $k = 9$), affect-enhancing variables ($r = 0.25$, 95% CI = [0.07, 0.43], $n = 557$, $k = 3$), goal-enhancing variables ($r = 0.23$, 95% CI = [0.13, 0.32], $n = 2,199$, $k = 17$), developmental/contextual enablers ($r = 0.21$, 95% CI = [0.08, 0.35], $n = 916$, $k = 6$) and goal-inhibiting variables ($r = 0.14$, 95% CI = [0.07, 0.21], $n = 2,398$, $k = 17$) all remained positively related to goal reengagement.

**Antecedents of goal-striving flexibility.** Regarding antecedents of goal-striving flexibility, the affect-enhancing variables category was significantly positively related to goal-striving flexibility, with the effect size being large. Optimism was the only individual variable in this category that was significantly related to goal-striving flexibility

**Table 1 | Meta-analytic results for antecedent variables**

| | Effect size (r) | 95% CI Lower | 95% CI Upper | s.e. | p | k | n | I² |
|---|---|---|---|---|---|---|---|---|
| **Disengagement** | | | | | | | | |
| Goal inhibiting | 0.136 | 0.048 | 0.227 | 0.045 | 0.003 | 58 | 10,240 | 96.544 |
| Agency inhibiting | 0.090 | −0.149 | 0.330 | 0.093 | 0.377 | 6 | 3,169 | 94.300 |
| Developmental/contextual barriers | 0.043 | −0.109 | 0.195 | 0.074 | 0.569 | 27 | 7,092 | 97.486 |
| Developmental/contextual enablers | 0.030 | −0.062 | 0.122 | 0.045 | 0.511 | 37 | 4,459 | 90.128 |
| Maladaptive personality traits | 0.020 | −0.084 | 0.124 | 0.050 | 0.694 | 23 | 2,357 | 80.896 |
| Agency enhancing | 0.007 | −.107 | 0.120 | 0.054 | 0.902 | 19 | 2,602 | 87.261 |
| Affect enhancing | <0.001 | −0.357 | 0.357 | 0.165 | >0.999 | 14 | 2,980 | 98.786 |
| Adaptive personality traits | −0.047 | −0.163 | 0.069 | 0.057 | 0.416 | 40 | 7,677 | 97.201 |
| Affect diminishing | −0.098 | −0.288 | 0.092 | 0.090 | 0.289 | 17 | 1,465 | 91.522 |
| Goal enhancing | −0.127 | −0.208 | −0.048 | 0.040 | 0.002 | 65 | 10,821 | 95.282 |
| **Reengagement** | | | | | | | | |
| Affect enhancing | 0.352 | 0.174 | 0.560 | 0.089 | 0.001 | 14 | 2,141 | 94.804 |
| Developmental/contextual enablers | 0.238 | 0.131 | 0.354 | 0.050 | 0.001 | 11 | 2,258 | 82.816 |
| Agency enhancing | 0.221 | 0.014 | 0.436 | 0.098 | 0.038 | 14 | 1,436 | 93.298 |
| Adaptive personality traits | 0.221 | 0.136 | 0.313 | 0.044 | <0.001 | 37 | 4,284 | 94.179 |
| Goal enhancing | 0.196 | 0.134 | 0.263 | 0.032 | <0.001 | 37 | 4,592 | 87.317 |
| Goal inhibiting | 0.111 | 0.026 | 0.198 | 0.042 | 0.012 | 35 | 4,168 | 87.798 |
| Agency linhibiting | 0.017 | −0.263 | 0.297 | 0.065 | 0.814 | 3 | 700 | 66.935 |
| Developmental/contextual barriers | −0.050 | −0.169 | 0.070 | 0.056 | 0.389 | 15 | 2,890 | 86.554 |
| Maladaptive personality traits | −0.091 | −0.231 | 0.048 | 0.059 | 0.166 | 8 | 1,060 | 80.725 |
| Affect diminishing | −0.114 | −0.315 | 0.087 | 0.091 | 0.238 | 12 | 1,103 | 88.418 |
| **Flexibility** | | | | | | | | |
| Affect enhancing | 0.560 | 0.476 | 0.791 | 0.067 | <0.001 | 8 | 1,547 | 86.535 |
| Goal enhancing | 0.305 | 0.017 | 0.612 | 0.121 | 0.041 | 7 | 1,433 | 95.291 |
| Adaptive personality traits | 0.233 | 0.100 | 0.375 | 0.065 | 0.002 | 17 | 2,478 | 94.489 |
| Developmental/contextual enablers | 0.169 | 0.086 | 0.255 | 0.041 | <0.001 | 27 | 8,431 | 94.534 |
| Agency enhancing | 0.154 | −0.070 | 0.381 | 0.088 | 0.136 | 6 | 369 | 83.995 |
| Goal inhibiting | 0.050 | −0.203 | 0.303 | 0.103 | 0.645 | 7 | 1,097 | 91.672 |
| Developmental/contextual barriers | 0.026 | −0.201 | 0.253 | 0.093 | 0.791 | 7 | 2,197 | 93.446 |
| Agency inhibiting | −0.149 | −0.362 | 0.062 | 0.067 | 0.109 | 4 | 369 | 66.772 |
| Maladaptive personality traits | −0.293 | −0.461 | −0.144 | 0.065 | 0.003 | 7 | 1,366 | 87.725 |
| Affect diminishing | −0.386 | −0.733 | −0.081 | 0.076 | 0.033 | 3 | 443 | 64.952 |

Significant two-sided meta-regression effects (p < 0.05) are highlighted. Effects are from independent analyses and not adjusted for multiple comparisons.

(r = 0.58, 95% CI = [0.49, 0.85], n = 1,547, k = 5). Adaptive personality traits were also positively associated with goal-striving flexibility, with the effect size being moderate. We were unable to meta-analyse individual variables associated with this category, as no single variable had been examined across more than one study. Developmental/contextual enablers evinced a small to moderate-sized positive association with goal-striving flexibility. None of the variables categorized under developmental/contextual enablers were individually significantly associated with goal-striving flexibility probably due to the small number of studies investigating each individual variable ($k_{range}$ = 2–3). Maladaptive personality traits and the affect-diminishing variables category were both significantly negatively related to goal-striving flexibility, with the effect sizes being large. There were too few individual variables measured across multiple studies to meaningfully meta-analyse individual effects in either category.

Sensitivity analyses considering only longitudinal and experimental studies produced significant associations between developmental/contextual enablers (r = 0.19, 95% CI = [0.04, 0.34], n = 1,343, k = 12)

and goal-striving flexibility, with a moderate effect size. Moreover, affect-enhancing variables (r = 0.48, 95% CI = [0.15, 0.89], n = 592, k = 2) were significant predictors of goal-striving flexibility, with a large effect size, although we synthesized this effect from a limited number of studies.

## Outcome variables

We present meta-regression coefficients, confidence intervals and heterogeneity estimates ($I^2$) for all outcome variables in Table 2. Heterogeneity was high for all outcome variables of goal disengagement, goal reengagement and goal-striving flexibility. In addition, we present moderator analyses for all outcomes in Supplementary Information 1.

**Goal disengagement outcome variables.** Goal disengagement was negatively related to goal progression, with the effect size being small to moderate. Breaking this category down into individual variables revealed a significant negative association between goal disengagement and goal progress/performance (r = −0.15, 95% CI = [−0.25, −0.05],

**Table 2 | Meta-analytic results for outcome variables**

|  | Effect size (*r*) | 95% CI Lower | 95% CI Upper | s.e. | *p* | *k* | *n* | *I²* |
|---|---|---|---|---|---|---|---|---|
| **Disengagement** | | | | | | | | |
| Impairment | 0.156 | 0.002 | 0.312 | 0.069 | 0.047 | 11 | 1,976 | 91.816 |
| Goal reduction | 0.128 | −0.012 | 0.269 | 0.066 | 0.071 | 17 | 3,080 | 94.211 |
| Wellbeing | −0.025 | −0.088 | 0.039 | 0.032 | 0.439 | 67 | 11,732 | 93.024 |
| Functioning | −0.055 | −0.115 | 0.004 | 0.028 | 0.067 | 19 | 3,486 | 81.654 |
| Illbeing | −0.118 | −0.171 | −0.065 | 0.027 | <0.001 | 123 | 18,569 | 91.786 |
| Goal progression | −0.128 | −0.209 | −0.048 | 0.040 | 0.002 | 41 | 6,039 | 89.969 |
| **Reengagement** | | | | | | | | |
| Wellbeing | 0.255 | 0.199 | 0.322 | 0.031 | <0.001 | 65 | 6,400 | 93.629 |
| Goal progression | 0.100 | −0.002 | 0.203 | 0.050 | 0.054 | 28 | 3,358 | 88.681 |
| Functioning | 0.097 | −0.117 | 0.311 | 0.101 | 0.353 | 18 | 2,391 | 96.492 |
| Goal reduction | 0.040 | −0.081 | 0.161 | 0.051 | 0.456 | 8 | 2,053 | 79.581 |
| Impairment | −0.149 | −0.308 | 0.008 | 0.072 | 0.060 | 12 | 2,107 | 92.182 |
| Illbeing | −0.206 | −0.252 | −0.166 | 0.022 | <0.001 | 104 | 12,617 | 81.492 |
| **Flexibility** | | | | | | | | |
| Wellbeing | 0.326 | 0.269 | 0.408 | 0.035 | <0.001 | 99 | 47,005 | 98.291 |
| Functioning | 0.188 | 0.048 | 0.333 | 0.070 | 0.010 | 39 | 21,488 | 99.123 |
| Goal progression | 0.156 | 0.060 | 0.253 | 0.047 | 0.003 | 25 | 6,636 | 92.007 |
| Goal reduction | −0.024 | −0.456 | 0.407 | 0.155 | 0.884 | 5 | 880 | 97.085 |
| Impairment | −0.144 | −0.421 | 0.131 | 0.113 | 0.246 | 7 | 670 | 88.823 |
| Illbeing | −0.266 | −0.364 | −0.182 | 0.045 | <0.001 | 58 | 29,552 | 98.007 |

Significant two-sided meta-regression effects (*p* < 0.05) are highlighted. Effects are from independent analyses and not adjusted for multiple comparisons.

*n* = 3,242, *k* = 14). Goal disengagement was also negatively related to illbeing, with the effect size being moderate. Examining the individual variables in this category indicated significant negative associations between goal disengagement and stress (*r* = −0.25, 95% CI = [−0.43, −0.08], *n* = 1,016, *k* = 7), anxiety (*r* = −0.21, 95% CI = [−0.29, −0.13], *n* = 2,446, *k* = 16) and depression (*r* = −0.14, 95% CI = [−0.21, −0.08], *n* = 9,027, *k* = 42). Finally, goal disengagement was positively related to impairment, with the effect sizes being moderate. However, we did not find any significant associations at the individual-variable level for either category. Sensitivity analyses considering only longitudinal and experimental studies supported small-to-moderate negative associations between goal disengagement and goal progression (*r* = −0.12, 95% CI = [−0.23, −0.003], *n* = 3,866, *k* = 24), as well as illbeing (*r* = −0.12, 95% CI = [−0.20, −0.04], *n* = 3,577, *k* = 34).

**Goal reengagement outcome variables.** Goal reengagement was negatively related to illbeing and positively related to wellbeing, both evincing moderate effect sizes. Next, we conducted analyses of individual variables comprising these categories. We obtained moderate-to-large positive associations between goal reengagement and the wellbeing variables: personal growth (*r* = 0.50, 95% CI = [0.27, 0.83], *n* = 450, *k* = 3), environmental mastery (*r* = 0.44, 95% CI = [0.28, 0.66], *n* = 450, *k* = 3), purpose in life (*r* = 0.29, 95% CI = [0.03, 0.56], *n* = 1,506, *k* = 8), positive affect (*r* = 0.30, 95% CI = [0.10, 0.52], *n* = 1,791, *k* = 9), self-acceptance (*r* = 0.28, 95% CI = [0.17, 0.41], *n* = 450, *k* = 3) and life satisfaction (*r* = 0.25, 95% CI = [0.21, 0.30], *n* = 3,141, *k* = 16). We synthesized most of these results from *k* ≤ 3 studies, except for positive affect, purpose in life and life satisfaction. We found associations between goal reengagement and the illbeing variables: stress (*r* = −0.32, 95% CI = [−0.47, −0.19], *n* = 812, *k* = 5), suicide ideation (*r* = −0.26, 95% CI = [−0.33, −0.20], *n* = 2,484, *k* = 4), depression (*r* = −0.24, 95% CI = [−0.31, −0.19], *n* = 4,953, *k* = 32), anxiety (*r* = −0.20, 95% CI = [−0.28, −0.12], *n* = 2,161, *k* = 16) and negative affect (*r* = −0.19, 95%

CI = [−0.32, −0.07], *n* = 1,424, *k* = 8). Sensitivity analyses including only longitudinal and experimental studies also supported the inverse pattern of associations between goal reengagement, illbeing (*r* = −0.17, 95% CI = [−0.24, −0.09], *n* = 2,487, *k* = 27) and wellbeing (*r* = 0.29, 95% CI = [0.21, 0.40], *n* = 1,391, *k* = 13).

**Goal-striving flexibility outcome variables.** Goal-striving flexibility was positively related to goal progression, with the effect size being moderate. An examination of individual variables in this category revealed a small positive association between goal-striving flexibility and persistence (*r* = 0.09, 95% CI = [0.02, 0.17], *n* = 3,130, *k* = 3), although it should be noted that this effect was synthesized from a small number of studies.

Goal-striving flexibility was also positively related to functioning and wellbeing, with the effect sizes being large. Analysis of individual variables comprising the functioning category indicated a large association between goal-striving flexibility and social relationships (*r* = 0.31, 95% CI = [0.12, 0.52], *n* = 1,943, *k* = 6), as well as physical quality of life (*r* = 0.25, 95% CI = [0.12, 0.41], *n* = 439, *k* = 4). Analyses of individual variables comprising the wellbeing category revealed moderate to large associations between goal-striving flexibility and psychological functioning (*r* = 0.46, 95% CI = [0.16, 0.85], *n* = 2,144, *k* = 3), positive subjective wellbeing (*r* = 0.41, 95% CI = [0.18, 0.69], *n* = 2,222, *k* = 9), purpose in life (*r* = 0.38, 95% CI = [0.12, 0.68], *n* = 1,704, *k* = 6), quality of life (psychological) (*r* = 0.38, 95% CI = [0.16, 0.65], *n* = 322, *k* = 3), life satisfaction (*r* = 0.36, 95% CI = [0.28, 0.48], *n* = 31,326, *k* = 17), quality of life (environmental) (*r* = 0.33, 95% CI = [0.10, 0.60], *n* = 322, *k* = 3), positive affect (*r* = 0.32, 95% CI = [0.15, 0.51], *n* = 18,562, *k* = 8), environmental mastery (*r* = 0.30, 95% CI = [0.13, 0.49], *n* = 1,075, *k* = 3), personal growth (*r* = 0.28, 95% CI = [0.00, 0.57], *n* = 1,075, *k* = 3) and autonomy (*r* = 0.26, 95% CI = [0.01, 0.51], *n* = 1,075, *k* = 3), noting that several of these effects were synthesized from a limited number of studies.

Finally, goal-striving flexibility was negatively related to illbeing, with the effect size also being large. Analysis of individual category variables indicated large negative associations with depression ($r = -0.39$, 95% CI = [$-0.48$, $-0.35$], $n = 6,420$, $k = 20$), anxiety ($r = -0.40$, 95% CI = [$-0.57$, $-0.27$], $n = 1,688$, $k = 8$) and negative affect ($r = -0.30$, 95% CI = [$-0.40$, $-0.21$], $n = 18,257$, $k = 7$). Sensitivity analysis including only longitudinal and experimental studies supported associations between goal-striving flexibility and goal progression ($r = 0.26$, 95% CI = [$0.09$, $0.45$], $n = 1,667$, $k = 9$), wellbeing ($r = 0.28$, 95% CI = [$0.17$, $0.40$], $n = 20,367$, $k = 19$) and illbeing ($r = -0.26$, 95% CI = [$-0.34$, $-0.19$], $n = 16,422$, $k = 8$).

### Study quality and bias

Average quality of included studies was high ($M = 0.81$, s.d. = 0.13). Further, ~36% of studies were longitudinal or experimental. Thus, although we observed a heavy reliance on cross-sectional and correlational methods in the field, over a third of studies provided a directional test of the associations proposed in our model. Sensitivity analyses that included only longitudinal and experimental studies largely mirrored our main analyses supporting the robustness of our conclusions. We used funnel plots and Egger's tests of funnel plot asymmetry to assess publication bias (Supplementary Table 1). The results indicated a substantive risk of publication bias in: (1) agency-inhibiting antecedents of goal disengagement, goal-enhancing antecedents of goal disengagement, affect-enhancing antecedents of goal disengagement; (2) all antecedents of goal reengagement excluding developmental barriers, maladaptive personality traits, agency-enhancing and inhibiting variables; (3) adaptive personality trait, affect-enhancing, developmental enablers and barriers, and goal-inhibiting antecedents of goal-striving flexibility; (4) goal progression, functioning, impairment and illbeing outcomes of disengagement; (5) wellbeing and illbeing outcomes of goal reengagement; and (6) wellbeing and illbeing outcomes of goal-striving flexibility. Despite the relatively high quality ratings of the included studies, we conclude that the overall standard of accumulated evidence remains low to moderate due to the reliance on cross-sectional studies, the risk of publication bias across many outcomes and antecedent categories, and the high degree of heterogeneity observed across most analyses. Although publication bias may have influenced the results of the current meta-analysis, we elected not to present bias-corrected estimates of effect sizes. Such corrections can convey a misleading sense of precision or certainty, given the lack of consensus on the most appropriate adjustment method and the sensitivity of different approaches to their underlying assumptions, which can yield markedly different results[40,41].

## Discussion

People often pursue goals that, for various reasons, become increasingly difficult or unattainable. Although extensive research has focused on persistence as a response to adversity or stress, the equally important and adaptive behaviour of goal adjustment has been largely overlooked. This imbalance in attention persists despite goal adjustment often being a more appropriate and beneficial response in many cases[2]. Goal adjustment manifests in various forms, as evidenced by numerous studies examining a wide range of predictors and consequences grounded in theoretical propositions[3,7,11]. Our objectives were to compile the antecedents and outcomes of goal adjustment, conduct a meta-analytic examination of their relations, and develop a conceptual model. This model captures both the types of variables that influence or result from different aspects of goal adjustment, as well as the strength and direction of these associations. The findings validated the role of most, although not all, proposed model constructs while revealing substantial variation in the strength and precision of associations synthesized from the literature. Overall, our model provides a roadmap for understanding the existing evidence on how aspects of the human experience relate to goal adjustment.

### Goal-striving flexibility

The goal-striving flexibility component of our model integrates research rooted in the dual process model[7,13]. The dual process model links development and goal regulation, proposing that goal striving flexibility becomes increasingly beneficial with age[42,43]. Accordingly, we observed a positive association between developmental/contextual variables that are typically beneficial to one's life course and goal-striving flexibility[44–46]. However, we found no evidence that events or circumstances typically adverse to lifespan development negatively impact goal-striving flexibility or any other goal adjustment outcome. It is possible that an environment supporting healthy development fosters exploratory coping mechanisms that promote flexibility in responding to goal-striving setbacks, yet negative life events do not necessarily hinder the development of such mechanisms. This aligns with the notion that diverse and enriching life experiences contribute to greater accommodative coping capacity[46,47].

Affect-enhancing variables and adaptive personality traits were also positively associated with goal-striving flexibility. Conversely, personality traits commonly considered as maladaptive, along with variables that undermine affect, negatively predicted goal-striving flexibility. Taken together, these findings indicate that goal-striving flexibility is more likely to emerge when individuals feel secure, exhibit stable regulation and possess emotional resilience. These capacities may help individuals manage the resources needed to influence their environment and achieve goals while providing a sense of security that allows them to confidently explore alternative options for goal pursuit. Moreover, personality traits and emotional regulation capabilities are likely key factors in an individual's ability to return to a healthy baseline after episodes of goal adjustment, which can be cognitively and emotionally demanding[38]. As a caveat to these findings, it is worth noting that when analyses were restricted to longitudinal and experimental studies, only developmental/contextual enablers and affect-enhancing variables emerged as significant predictors of goal-striving flexibility. Although this pattern may raise questions about the temporal precedence and predictive validity of other proposed antecedents, we argue that these null effects are likely attributable to both the limited number of longitudinal and experimental investigations, and the extended developmental timeline often required for flexible goal-adjustment capacities to manifest[15]. Addressing the directionality and complexity of associations among antecedents, goal-adjustment capacities and outcomes may be best accomplished through lifetime longitudinal or cohort studies, coupled with advanced statistical techniques that account for interdependencies over time, such as dynamic structural equation modelling[48].

A core tenet of the dual process model is that maintaining a flexible approach to goal striving fosters a more fulfilling and healthier life[7,13]. Replicating previous work[5], we obtained strong support for associations between goal-striving flexibility and indicators of wellbeing and functioning. Our meta-analytic results revealed that goal-striving flexibility enhances both wellbeing and functioning while protecting against illbeing, highlighting its multifaceted role in promoting health and wellbeing. The benefits of maintaining a flexible disposition toward goal adjustment appear to be widespread, with positive implications for mental health, physical health and social functioning. We also extended previous findings by demonstrating a positive association between flexible goal adjustment and goal attainment. This aligns with the proposition that maintaining flexibility is crucial, given the cyclic nature of goal setting, striving and attainment/adjustment throughout life[15].

### Disengagement and reengagement

Theoretical accounts of goal-adjustment initiation have primarily focused on an individual's internal state and its relation to their goal. For example, both the Action Crisis model[34] and control-inspired theories[9,11] highlight how failing to achieve adequate progress with

a goal engenders negative emotionality, which in turn conduces to disengagement. Taking this proposition a step further, some authors have suggested that highly negative states, such as depression, are adaptive insofar as they facilitate goal disengagement[49,50]. Our results provided some support for these suggestions. We found that events or states that were non-conducive to goal striving, in particular experiencing an action crisis, threat appraisals and receiving negative feedback about one's goals were positively related to goal disengagement, while variables conducive to goal striving were negatively related to disengagement. Affective variables, specifically optimism and positive affect, were related to goal reengagement. However, we failed to find sufficient evidence that affect played a role as an antecedent of goal disengagement. The Action Crisis model emphasizes that negative emotional reactions to setbacks prompt individuals to weigh the costs and benefits of continuing their goal pursuit[51]. Thus, it is possible that negative emotionality is not directly linked to disengagement, but instead moderates or mediates other goal-related cognitive processes, such as goal appraisals[52] and cost–benefit thinking[51], which in turn facilitate adjustment. Moreover, at the later stages of the goal-striving cycle, our findings indicated that goal disengagement was associated with improvements in several indicators of psychological illbeing, specifically reductions in depression, stress and anxiety. These patterns suggest that disengagement may function as a psychological 'pressure release valve' allowing individuals to alleviate their emotional burden associated with unattainable goals. By facilitating a return to a more regulated emotional state, disengagement may also serve as preparatory mechanism that enables subsequent processes, such as goal reengagement.

Several variable categories predicted goal reengagement. One of the more counterintuitive findings was that both antecedents that promote goal striving and those that impede it are positively related to reengagement. Goal reengagement encompasses both the continuation of goal striving at a high level of one's goal hierarchy and the discontinuation and subsequent adjustment of striving at lower levels[3]. Viewed in this light, it follows that both processes involved in the cessation of old modes of goal striving and processes that reinforce striving for the goal at a higher conceptual level contribute to goal reengagement. To illustrate this point, consider the impact of an action crisis on goal-striving processes. On the one hand, the experience of an action crisis may lead to reduced commitment and effort toward the original goal[34], decreasing the likelihood that the individual will reengage with that same goal in the future, potentially due to processes such as goal devaluation[53]. In this context, a researcher assessing reengagement with the original goal may observe a negative association between reengagement and indicators of commitment or effort. On the other hand, an action crisis can also prompt disengagement from the unattainable goal and facilitate a shift in focus toward a new, more attainable goal[54]. Consequently, a researcher examining reengagement with alternative goals would probably find a positive association between reengagement and commitment or effort. These divergent patterns highlight the importance of specifying the target of reengagement (that is, original vs new goals) when interpreting findings on goal adjustment processes. We propose that processes that either facilitate or impede goal striving will differentially predict disengagement and reengagement, depending on the hierarchical level at which a goal is examined. For example, we would expect research examining goal-pursuit activities at the micro level (that is, current goal-striving activities) to identify associations with goal-striving inhibition, but research examining goal pursuit at the macro level (that is, commitment to a higher-order goal) to identify associations between processes that enhance goal striving. Interactions across these levels may explain how goal inhibitory processes can positively predict reengagement, potentially through their impact on disengagement[55].

Affect, personality, development/context and agency also predicted reengagement. Taken together, these findings align with the idea that goal reengagement occurs when individuals remain highly committed to their goal and have both the personal and environmental resources needed to pursue it effectively[3]. Importantly, many associations described here were present when we limited our analysis to experimental and longitudinal work only, providing initial evidence that these factors may chronologically precede or predict goal disengagement and reengagement.

Turning to outcomes, goal disengagement negatively predicted goal progression, supporting the idea that its primary function is to decelerate ongoing goal striving[56], thus creating space for adopting alternative approaches to the goal. We did not observe significant associations between reengagement and goal-striving outcomes, despite theoretical propositions to the contrary[20,57]. The quantification of associations between reengagement and goal-related outcomes may have been complicated by variations in how goal-related outcomes are operationalized (for example, measuring progress for the original vs reengaged goal), when reengagement capacity is assessed (for example, before or after goal adjustment), and whether objective or subjective reports of goal outcomes are used[58]. Again, we make the case that the relation between reengagement and goal-striving outcomes depends on the level at which the goal is analysed. At the micro level, goal striving may show a negative or null association between reengagement and attainment of the original goal. However, at the macro level, reengagement may be positively linked to attainment of the overarching goal.

We obtained evidence for associations between disengagement and reengagement and wellbeing/illbeing outcomes, replicating previous findings[4]. Importantly, our results support the theoretical proposition that successful disengagement and reengagement can be distinctly beneficial[11]. As mentioned, disengagement may help to protect individuals from a deteriorating state caused by continued goal-striving failure, but it does not actively promote positive states. Accordingly, we found disengagement to be negatively associated with illbeing, but not with wellbeing. Conversely, reengagement should both reduce illbeing and enhance wellbeing by creating ongoing opportunities to benefit from sustained goal pursuit. This was demonstrated in our work through negative associations between reengagement and illbeing, and positive associations between reengagement and wellbeing. In summary, disengagement and reengagement serve distinct yet complimentary roles in promoting wellbeing.

We found little evidence linking goal disengagement or reengagement to better functioning. One explanation is that improvements in functioning require time to manifest and may only become apparent after individuals achieve success with newly adopted goals. This interpretation aligns with our finding that dispositional flexibility, rather than more proximal disengagement or reengagement, more strongly predicts functioning. Notably, we observed a positive association between disengagement and impairment. Although this could reflect a 'dark side' of disengagement—where letting go of goals offers short-term relief but risks longer-term purposelessness and dysfunction[11]—this pattern was not evident in longitudinal or experimental studies. An alternative explanation is that the association is bidirectional, with impairment potentially prompting disengagement as a reactive strategy. Given these complexities, we advice caution in interpreting this finding and highlight the need for further research to disentangle the temporal and causal dynamic underlying the relationship between disengagement and impairment.

## Future directions
**Goal adjustment as an emergent phenomenon.** Given the variety of antecedents linked to goal adjustment, it is unlikely that the onset of adjustment can be attributed to any single variable. Moving beyond conceptualizing goal adjustment as a homogeneous and sequential process, such as Klinger's[59] incentive–disengagement cycle, we propose that antecedents, as categorized in the present review, probably

operate in an additive or synergistic manner rather than in isolation. Under this proposal, transitions between phases of active goal pursuit and goal adjustment emerge when the cumulative (or multiplicative) activation of interdependent antecedents reach a tipping point in an individual's goal striving[60]. The amount of activation needed to reach a tipping point may be influenced or moderated by the individual's degree of goal-striving flexibility. Future research could explore combinations of antecedents as predictors of disengagement and reengagement, and how these relations are influenced by a person's overall tendency toward flexible goal striving. Our model may assist researchers in deciding which variables to examine. To offer a more comprehensive account of goal adjustment, we suggest that examining interactions between variables in different model categories will be more informative than focusing on variables within the same category. For example, a researcher might explore how affect influences goal reengagement across different goal-striving contexts, or whether personality traits influence decisions to adjust goals when perceived agency is high versus low.

**Filling evidence gaps.** Although we attempted to simplify antecedents and outcomes of goal adjustment into broad descriptive categories, we acknowledge substantial heterogeneity in the literature. By consolidating the evidence, we intended to provide researchers with a clear overview of existing gaps, with our integrative framework serving as a guide for a more coordinated and focused research agenda. Follow-up investigations could assess the comprehensiveness and validity of our proposed categories. Our model also implies sequential paths from antecedents to outcomes via goal adjustment processes. Examining these paths will be a critical test of our model. Further, we determined antecedent categories by compiling the proposed antecedents identified in the literature. However, much of this literature is cross sectional, which limits the ability to make claims about the directionality of relations between variables. Several, although not all, associations persisted when analyses were restricted to longitudinal and experimental studies, offering preliminary evidence that at least some antecedent and outcome categories may follow the proposed temporal sequence. However, testing the assumptions of our model will require a combination of cross-sectional, longitudinal and experimental evidence. In addition, methods such as meta-analytic structural equation modelling could eventually be used to examine our model in its entirety.

**Testing goals across multiple levels and multiple goals.** Our findings regarding goal facilitatory and inhibitory antecedents indicate that the relation between antecedents and disengagement or reengagement depends on whether goal striving is examined at the micro or macro level. This implies that a multilevel approach to studying goal striving may be essential[9]. A key limitation in the current literature is the lack of clarity regarding the level of the goal hierarchy at which disengagement and reengagement are assessed. Adjustment processes may differ markedly depending on whether they occur within the same hierarchical level (for example, shifting from regular running to walking) or across levels (for example, from exercising to dieting to achieve weight loss). These distinctions are consequential, as disengagement may hinder progress at one level, whereas reengagement can preserve or redirect progress toward higher-order goals. For example, experiments that manipulate goal attainability have demonstrated that individuals often make greater progress toward overarching aims when they relinquish unattainable lower-order goals[22,23,61]. Moreover, the boundaries between goal progress, disengagement and reengagement are often more fluid than typically presented. An individual may, for instance, appear to abandon the goal of finding a romantic partner and redirect effort toward self-improvement, yet may not have fully disengaged from the original goal even after months or years of non-pursuit. This conceptual ambiguity is reflected in emerging evidence on goal shelving and freezing phenomena[62,63]. We contend that a more nuanced understanding of goal adjustment requires researchers to begin by precisely specifying the referent of disengagement and reengagement, namely, disengagement or reengagement with what?

Further, goals are rarely pursued in isolation, and decisions to adjust goal pursuit can be driven by competition from competing goals[64]. A recent framework designed to conceptualize goal disengagement within the context of multiple goals has reached conclusions similar to those presented in the current review[65]. Specifically, the authors also identified a range of personal and contextual variables that can both encourage and discourage disengagement, emphasized the importance of considering goal hierarchies, and acknowledged that goal adjustment is a dynamic process influenced by multiple variables rather than a discrete event. Future multilevel approaches to examining goal adjustment should be expanded to consider not only goal hierarchies but also intergoal dynamics[55].

**Adjustment as part of the bigger picture.** We suggest that a comprehensive understanding of goal adjustment be integrated into a broader explanation of goal striving that encompasses other goal-regulatory processes, such as persistence. Integrative goal pursuit models, such as those derived from control theory[10], often position adjustment as a reaction to failed goal regulation. Our work indicates that decisions to persist or adjust goals are more nuanced, depending on the individual, their environment and the goal itself. Researchers could explore the conditions under which combinations of antecedents differentially predict persistence versus adjustment, and examine when these two goal-regulatory strategies lead to adaptive versus maladaptive outcomes for goal striving, wellbeing and functioning.

### Strengths and limitations

Our model addresses an empirical lacuna in the literature, providing a unified perspective on goal adjustment. Furthermore, our meta-analytic efforts offer a comprehensive overview of the variables that influence or result from goal adjustment. Previous meta-analyses of goal adjustment have considered a single outcome (for example, wellbeing[4,5]), focused on a single way of measuring goal adjustment (for example, Goal Adjustment Scale[4]), only examined adjustment for a specific life domain (for example, people with long-term health conditions[6]), and did not analyse antecedent variables. We adopted an inclusive approach by taking into account various methods of measuring goal adjustment across all life domains. In addition, we expanded on previous evidence syntheses by investigating the consequences of goal adjustment for both functioning and goal striving itself.

Our work has several limitations. We acknowledge that our categorization process involves a degree of subjectivity and that our ability to draw conclusions about the directionality of some relations is limited, given that most effects are derived from cross-sectional studies. Nevertheless, for those studies, the operationalization of variables as antecedents or consequences of goal flexibility was grounded in theoretical reasoning and in many cases supported by sensitivity analyses of longitudinal and experimental studies. A related limitation concerns the subjective nature of the variable categorization process. Categorization decisions were reached through consensus across multiple researchers with expertise spanning diverse areas of psychology and were guided by the theoretical framing of variables in the primary literature. The classification of antecedents and outcomes as positive or negative exemplars within overarching categories followed the same approach. Again, these decisions were primarily informed by the theoretical positioning adopted by original study authors. Nonetheless, we acknowledge that alternative classifications are plausible, and that some variables may reasonably fit within multiple or differing categories. These possibilities warrant empirical scrutiny.

Further, we acknowledge that combining a wide range of variables into broad categories introduced considerable heterogeneity. An alternative approach would be to meta-analyse each variable

independently, which we also did and summarized in text, with full details of the analysis in Supplementary Information 1. As the purpose of our meta-analytic effort was to provide an overview of the state of the field to inform future research, we contend that our current analysis strikes an optimal balance between usability and sensitivity.

### Constraints on generality

The processes underlying goal adjustment are heterogeneous across the lifespan[15] and vary between cultures[66]. Given that much of the relevant research has been conducted with samples from Western, educated, industrialized and democratic societies[67], we have concerns regarding the generalizability of the existing findings. There is a need to diversify this field by examining how the processes underpinning goal adjustment differ both between and within cultures. While the majority of research has been published in English, we note that we also included several non-English articles. Although we are confident that we have captured the key findings in the literature, we also acknowledge the potential English language bias, both in our own work and in the field.

The study of goal adjustment has only gained notable traction in recent years and has been pursued by a relatively diverse group of researchers. Most articles have been published by authors from Western Europe and North America, with the most prolific and seminal scholars in the field also hailing from these regions. However, there have been notable contributions from researchers around the world, and the scholarship on goal adjustment does not appear to be dominated by a single author, institution or research group.

## Conclusion

We sought to consolidate the fragmented body of literature on goal adjustment. Our findings corroborated previous research highlighting the benefits of goal adjustment for wellbeing. In addition, we identified several unique and shared predictors of flexible goal adjustment, disengagement and reengagement. Our conceptual model, which is informed by both theory and empirical findings, generates testable hypotheses regarding the organization of antecedent and outcome variables into broader categories, as well as the interrelations among antecedent categories, goal adjustment processes and outcome categories. The model provides a framework that can steer a more cohesive and methodical exploration of this emerging field.

## Methods

### Transparency and openness

We adhered to the Transparency and Openness Promotion guidelines[68]. We preregistered the meta-analytic protocol on the Open Science Framework (https://osf.io/9hmw8) using the Preferred Reporting Items for Systematic Reviews and Meta-Analyses-Protocol template (PRISMA-P[69]). We note four deviations from the preregistered protocol. First, we employed the QualSyst tool[70] to assess study quality for all included studies, in contrast to our initial intention to employ the Quality of Survey Studies in Psychology tool[71] for correlational studies and QualSyst for experimental studies. QualSyst is appropriate for both correlational and experimental studies; hence, we made this change to maintain consistency in the study quality rating instrument across studies and facilitate coding efficiency. The second deviation involved categorizing antecedent and outcome variables, a decision informed by the Cochrane guidelines on handling sparse data[72]. When examining uncategorized antecedents, we located multiple effects with $k \leq 2$ effect sizes in the extracted data. Effect size estimates, heterogeneity estimates and moderator analyses are less reliable and informative when there is a limited number of effects to synthesize. Moreover, although antecedent variables examined in the literature are distinct, there is a degree of overlap in many concepts, and variables often coalesce under broader categories; for example, motivation is a common umbrella category in psychology that encompasses a range of distinct yet related variables. A similar methodological approach

has been employed in other areas of psychology[73] and by ref. 4 in their meta-analysis of wellbeing outcomes associated with goal disengagement and reengagement. A further methodological deviation in the present review was the inclusion of articles not written in English. We made this decision to minimize potential publication language biases and enhance the comprehensiveness of the review. The authorship team is proficient in English, Danish, French, German, Greek, Hindi and Tamil. We translated one article published in Japanese, a language not spoken by any team member, using a large language model (ChatGPT, version 4o)[74]. Finally, we did not solicit unpublished manuscripts from the listserv SPSP Connect! as outlined in our protocol, because this service required paid membership.

### Literature search

The first author conducted a systematic search—initially on 25 May 2022, with updates on 15 July 2023 and again on 22 May 2025—of the following electronic databases: Web of Science, Scopus, PsycInfo, Business Source Ultimate, Proquest Dissertations and Theses Global, and Medline. We did not impose limits on the earliest or latest publication dates of articles. We searched for the key term "goal" with a 4-word adjacency search with the terms: ("reengage*" OR "disengage*" OR "unattain*" OR "dis-engag*" OR "re-engag*" OR "give up" OR "adjust*" OR "flexib*"). We list specific search terms used for each database in the preregistration and in Supplementary Information 2. We supplemented our main search with an ascendancy/descendancy search, assisted by the Paperfetcher web app[75].

### Eligibility criteria

Studies were eligible if they measured goal adjustment in any form (for example, self-reported adjustment capacity, objectively assessed behavioural adjustment) and either one or all of the following: (1) antecedents predicting goal adjustment; (2) goal-related outcomes; (3) functional outcomes; and (4) wellbeing outcomes. We included studies if they provided adequate information to compute an effect size (from either correlations or mean differences) or we could obtain this information directly from the authors through personal contact.

We excluded studies if: (1) the article's full-text was unavailable from university library subscriptions, via ResearchGate, through the university's institutional document request service, or via direct correspondence with the author; (2) the information necessary to compute an effect size was unavailable in the full-text document or following direct contact with the author; (3) the results were from a conference abstract rather than a full-text; or (4) the study tested goal persistence but not adjustment. We also excluded studies if (5) they examined goal adjustment in children younger than 10 years old. Young children are still developing self-regulatory capacities and have limited control over their goals, or control is jointly determined with parents, complicating their assessment of goal regulation. Finally, we excluded studies (6) when it was impossible to disentangle goal adjustment from other cognitive or behavioural processes. A common example of (6) is the Optimization of Primary and Secondary Control Scale of ref. 76, in which the disengagement subscale is often combined with subscales measuring other control processes (for example, social and intra-individual comparisons, self-protective attributions) that fall outside the scope of our definitions of adjustment processes. Thus, we only included studies in which it was possible to disentangle scores for goal-adjustment relevant subscales or items from other non-relevant subscales or items.

### Article screening

We exported articles retrieved through the literature search to EndNote (v.20) and removed duplicates. We used Research Screener[77] to semi-automate the abstract screening process. Systematic reviewers are highly likely to identify all eligible articles by screening 5–35% of the total pool using Research Screener. On average, all relevant articles

are identified after screening 12.8% of articles or after 20 rounds (50 articles per round) of screening[77]. The first author continued screening articles until 15 consecutive rounds passed without identifying any new eligible articles. As a result, the first author manually screened 33% of eligible articles. The first and third authors reviewed all full texts flagged for potential inclusion and resolved disagreements through discussion.

### Data extraction
The first, third, fourth and fifth authors extracted all data items from eligible studies using a predetermined data extraction form available on the project's OSF page. We ensured the accuracy and consistency of the extracted data by having each of the four coders randomly review 20% of the data extraction forms completed by the other coders. We did not track the level of agreement in data extraction and study quality coding but note that all individuals involved in these processes were experienced researchers who had authored meta-analyses and engaged in data extraction and study quality coding previously. Although there were minor discrepancies in data extraction and study quality, these were straightforwardly resolved via discussion.

We included data from cross-sectional, longitudinal, experimental and diary studies. To maximize the likelihood of capturing effects of goal adjustment on outcomes in longitudinal studies, we extracted data for the latest timepoint reported for each variable in the study. For example, if a study measured goal adjustment at baseline only and measured the outcome both at baseline and a second timepoint, we coded the correlation between adjustment at baseline and the outcome at time two. For diary studies, we extracted only between-person data to maintain consistency with the between-person data obtained from the rest of the studies.

### Study quality
The first and fourth author evaluated the quality of all included studies using the QualSyst tool. It comprises 14 standardized items that appraise design quality and is appropriate for all research designs included in this meta-analysis[70]. The authors of QualSyst identified conservative (<0.75) and liberal (<0.55) thresholds of minimum quality ratings for study inclusion in reviews. Although we did not exclude studies based on quality ratings, we adapted these thresholds as cut-off points for identifying high (≥0.75), moderate (0.74–0.56) and low (≤0.55) quality studies.

### Measures
**Goal disengagement.** We defined goal disengagement as the behavioural cessation of goal pursuit/persistence or as self-reported cognitive disengagement from goal striving. Disengagement may occur in isolation and be followed by reengagement or another form of adjustment. We only coded effect sizes when it was possible to disentangle antecedents and outcomes of goal disengagement from antecedents and outcomes of other adjustment processes.

**Goal reengagement.** We defined goal reengagement as the behavioural resumption of pursuit with a new or modified goal, or self-reported cognitions toward reengaging. Again, we only coded effect sizes when it was possible to disentangle antecedents and outcomes of goal reengagement from antecedents and outcomes of other adjustment processes.

**Flexible goal adjustment.** We defined flexible goal adjustment as an individual's general propensity toward flexible adjustment of goal striving or accommodative goal striving. Studies that measured this construct did not disentangle separate processes of goal adjustment (for example, disengagement, reengagement). Where we could identify distinct disengagement and reengagement subprocesses, we coded them separately as disengagement and reengagement.

Where disengagement and reengagement were combined or were not clearly separable within scale items, we coded them as flexible goal adjustment.

**Antecedents.** We defined antecedents as events, cognitions or emotional states that temporally preceded (that is, in longitudinal studies) goal adjustment, or were hypothesized by the original study's authors to predict goal adjustment.

**Goal-related outcomes.** We defined goal-related outcomes as self-reported or objectively measured outcomes that: (1) directly indicated pursuit of goals (for example, goal progress, effort) or goal-directed attachment/cognition (for example, goal commitment, rumination on goals) and (2) temporally followed, or were hypothesized by the original authors to be predicted by, goal adjustment.

**Functioning outcomes.** We defined functioning outcomes as those related to an individual's physical (for example, physiological health/impairment), social (for example, interpersonal relationships) or societal (for example, ability to function in society, such as at work) effectiveness that temporally followed, or were hypothesized by the original authors to be predicted by, goal adjustment.

**Wellbeing outcomes.** We defined wellbeing outcomes as those related to an individual's mental health, that temporally followed, or were hypothesized by the original authors to be predicted, by goal adjustment.

**Additional variables.** We also extracted information on: whether studies were preregistered; type of study design; life domain in which the goal was pursued; whether the study examined goal striving in a general population or a specific one (for example, individuals with a particular medical condition); participant mean age; proportion of female participants; country of sample's origin; language spoken by participants; whether the study tested specific or general goals; whether goals were assigned by the researcher or set by the participants; and whether the original goal attainability was manipulated.

### Conceptual model development and data analysis
We developed our integrative model of goal adjustment both inductively, drawing from the data collected in our literature search, and deductively, on the basis of meta-analytic associations identified in the literature. To develop our model, the first and last authors, both of whom are experts on goal adjustment, first reviewed all extracted antecedent variables and developed thematic categories that aligned with well-established but broad psychological constructs (for example, personality, affect, agency), processes (for example, goal striving) and contexts (for example, development/environmental context). The authors then collaboratively assigned individual antecedent variables into these categories. The authors further divided these antecedents on the basis of whether primary studies and the broader psychological literature views them as enhancing/adaptive or inhibitory/maladaptive, culminating in the following categories: goal-enhancing/inhibiting variables; affect enhancing/diminishing variables; adaptive/maladaptive personality traits; developmental and contextual barriers/enablers; and agency-enhancing/inhibiting variables. We refined the categorization of variables through an iterative process that involved consulting primary studies and examining the theoretical frameworks used by original authors to contextualize their variables. Final categorization decisions were reviewed and approved by all manuscript authors. We similarly grouped outcomes into six overarching categories, conceptualized as three opposing pairs: wellbeing versus illbeing, functioning versus impairment, and goal progression versus goal reduction. We present a detailed mapping of individual variables to model constructs in Supplementary Table 4,

as well as a summary of the variables assessed across primary studies in Supplementary Table 5.

Next, we conducted separate meta-regressions between antecedent categories and goal adjustment capacities, as well as meta-regressions between goal adjustment capacities and outcome categories. For each meta-analysis, we converted bivariate correlations to Fisher's $z$-scores. For studies employing experimental methodologies or comparing mean between-group differences, we calculated Hedge's $g$ as a standardized mean difference, then converted it to Fisher's $z$-scores for analysis and $r$ values for reporting, using formulas in ref. 78. Where means and standard deviations were unreported for group differences but $F$ statistics were available, we imputed effect sizes[78]. To interpret effect sizes, we used the benchmarks of 0.11, 0.19 and 0.29, which correspond to small, medium and large effect sizes, respectively, that are typically reported in psychology studies[24]. We conducted all data analyses in R (v.4.5)[79] using the metafor package (v.4.8)[80].

Given that multiple effects derived from the same study cannot be considered independent, we combined effect sizes employing three-level random effects models, which allow for the separation of variance for each effect size (level 1), variance between effect sizes of the same study (level 2), and variance between studies (level 3)[81]. We determined statistical heterogeneity using $I^2$, which indicates the proportion of total variance in effect estimates that results from heterogeneity rather than sampling error. For the $I^2$ statistic, the following cut-off points are generally used to interpret heterogeneity: 0%–40% = might not be important; 30%–60% = may represent moderate heterogeneity; 50%–90% = may represent substantial heterogeneity; and 75%–100% = considerable heterogeneity[82].

Where possible, we examined the potential moderating influence of several study characteristics: average sample age, proportion of female participants, whether the sample was drawn from a specialized (for example, clinical, student, professional) population, the country of origin (that is, geographic region of the sample) and study quality ratings. We present a summary of key findings from these moderator analyses in Supplementary Information 1, with full analytical outputs available on the project's OSF page. In addition, for each primary analysis, we conducted a sensitivity analysis restricted to longitudinal and experimental studies. This approach allowed for a more stringent test of the theoretical proposition that antecedents predict goal adjustment, and that goal adjustment, in turn, predicts outcome variables.

We assessed publication bias via sunset enhanced funnel plots and Egger's test of funnel plot asymmetry (metaviz package, v.0.99.5)[83]. Finally, we meta-analysed the associations between goal adjustment capacities and individual antecedent and outcome variables (that is, excluding our model groupings) to examine the size and direction of relation between specific individual variables studied in the literature and goal adjustment capacities. We report key findings from these analyses in text and the full analyses in Supplementary Tables 1, 3 and 4.

### Reporting summary
Further information on research design is available in the Nature Portfolio Reporting Summary linked to this article.

## Data availability
All data for this project have been made publicly accessible via the Open Science Framework at https://osf.io/6qft4/ (ref. 84). Data can be accessed directly at https://osf.io/6qft4/files/osfstorage/685bc7dda251d5640d5cfad3.

## Code availability
All analysis scripts for this project have been made publicly accessible via the Open Science Framework[84]. The main analysis can be accessed directly at https://osf.io/6qft4/files/osfstorage/685bc7dd15a516490ea1c8ac and moderator analysis can be accessed directly at https://osf.io/6qft4/files/osfstorage/685bc7e06f77ed49ab0705d5.

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

## Acknowledgements

This project was funded by an Australian Research Council grant (DP200101555) awarded to some of the authors (principal investigator: N.N.; co-investigators: C.S., D.F.G., B.J., C.T.-N.). The funders had no role in study design, data collection and analysis, decision to publish, or preparation of the manuscript.

## Author contributions

H.R. helped develop the study, conducted the searches, screened studies, extracted data, conducted analysis and wrote the manuscript. C.S. made substantial contributions to writing the manuscript and helping develop the conceptual model. H.S., P.W. and S.M. assisted with screening and data extraction. B.J., C.T.-N. and D.F.G. assisted with developing the study and drafting the manuscript. D.F.G. also supported the data analysis. N.N. obtained funding for the work, helped develop the study and provided significant input into the development of the conceptual model and writing the manuscript. All authors contributed their feedback on the variable categorization scheme for the conceptual model.

## Competing interests

The authors declare no competing interests.

## Additional information

**Correspondence and requests for materials** should be addressed to Nikos Ntoumanis.

Hugh Riddell [1,2], Constantine Sedikides [3], Hamsini Sivaramakrishnan [1,4], Phoebe Wan [5], Silvio Maltagliati [6,7], Ben Jackson [4,8], Cecilie Thøgersen-Ntoumani [9,10], Daniel F. Gucciardi [5] & Nikos Ntoumanis [9,10] ✉

[1]Curtin School of Population Health, Curtin University, Perth, Western Australia, Australia. [2]enAble Institute, Perth, Western Australia, Australia. [3]Center for Research on Self and Identity, School of Psychology, University of Southampton, Southampton, UK. [4]The Kids Research Institute, Perth, Western Australia, Australia. [5]Curtin School of Allied Health, Curtin University, Perth, Western Australia, Australia. [6]University of Grenoble Alpes, SENS, Grenoble, France. [7]Human and Evolutionary Biology Section, Department of Biological Sciences, University of Southern California, Los Angeles, CA, USA. [8]School of Human Sciences (Exercise and Sport Science), University of Western Australia, Crawley, Western Australia, Australia. [9]Danish Centre for Motivation and Behavior Science, University of Southern Denmark, Sønderborg, Denmark. [10]School of Sport, Exercise and Rehabilitation Sciences, University of Birmingham, Birmingham, UK. ✉e-mail: n.ntoumanis@bham.ac.uk

# Reporting Summary

## Statistics

For all statistical analyses, confirm that the following items are present in the figure legend, table legend, main text, or Methods section.

| n/a | Confirmed | |
|---|---|---|
| ☐ | ☒ | The exact sample size (*n*) for each experimental group/condition, given as a discrete number and unit of measurement |
| ☒ | ☐ | A statement on whether measurements were taken from distinct samples or whether the same sample was measured repeatedly |
| ☐ | ☒ | The statistical test(s) used AND whether they are one- or two-sided<br>*Only common tests should be described solely by name; describe more complex techniques in the Methods section.* |
| ☐ | ☒ | A description of all covariates tested |
| ☐ | ☒ | A description of any assumptions or corrections, such as tests of normality and adjustment for multiple comparisons |
| ☐ | ☒ | A full description of the statistical parameters including central tendency (e.g. means) or other basic estimates (e.g. regression coefficient) AND variation (e.g. standard deviation) or associated estimates of uncertainty (e.g. confidence intervals) |
| ☐ | ☒ | For null hypothesis testing, the test statistic (e.g. *F*, *t*, *r*) with confidence intervals, effect sizes, degrees of freedom and *P* value noted<br>*Give P values as exact values whenever suitable.* |
| ☒ | ☐ | For Bayesian analysis, information on the choice of priors and Markov chain Monte Carlo settings |
| ☐ | ☒ | For hierarchical and complex designs, identification of the appropriate level for tests and full reporting of outcomes |
| ☐ | ☒ | Estimates of effect sizes (e.g. Cohen's *d*, Pearson's *r*), indicating how they were calculated |

*Our web collection on statistics for biologists contains articles on many of the points above.*

## Software and code

Policy information about availability of computer code

| | |
|---|---|
| Data collection | Endnote was use to collate articles identified through the systematic search |
| Data analysis | All analysis was We conducted all data analyses in R (Version 4.5; R Core Team, 2022) using the metafor (Veichtbauer, 2010) and metaviz packages (Kossmeier et al., 2020). All analysis code has been published on the project's OSF page (https://osf.io/6qft4/). |

For manuscripts utilizing custom algorithms or software that are central to the research but not yet described in published literature, software must be made available to editors and reviewers. We strongly encourage code deposition in a community repository (e.g. GitHub). See the Nature Portfolio guidelines for submitting code & software for further information.

## Data

Policy information about availability of data

All manuscripts must include a data availability statement. This statement should provide the following information, where applicable:
- Accession codes, unique identifiers, or web links for publicly available datasets
- A description of any restrictions on data availability
- For clinical datasets or third party data, please ensure that the statement adheres to our policy

All data and analysis scripts for this project have been made publicly accessible via the Open Science Framework (https://osf.io/6qft4/).

## Research involving human participants, their data, or biological material

Policy information about studies with human participants or human data. See also policy information about sex, gender (identity/presentation), and sexual orientation and race, ethnicity and racism.

| | |
|---|---|
| Reporting on sex and gender | We included participant gender distribution (% female participants) as a moderator in our analyses. Most included studies did not explicitly describe biological sex of participants, therefore we were unable to conduct analyses pertaining to biological sex and operate under the assumption that gender reported in the included studies reflects identified gender not biological sex. |
| Reporting on race, ethnicity, or other socially relevant groupings | We analyzed the country in which study samples were drawn from as a moderator, but did not include further information about the explicit race or ethnicity of participants within these samples, as this information was not clear in from the majority of included manuscripts. |
| Population characteristics | In addition to the above mentioned population characteristics, we included participant age and participant subsample as moderators in our analyses |
| Recruitment | N/A |
| Ethics oversight | N/A |

Note that full information on the approval of the study protocol must also be provided in the manuscript.

# Field-specific reporting

Please select the one below that is the best fit for your research. If you are not sure, read the appropriate sections before making your selection.

☐ Life sciences  ☒ Behavioural & social sciences  ☐ Ecological, evolutionary & environmental sciences

For a reference copy of the document with all sections, see nature.com/documents/nr-reporting-summary-flat.pdf

# Behavioural & social sciences study design

All studies must disclose on these points even when the disclosure is negative.

| | |
|---|---|
| Study description | Meta-analysis of qualitative studies |
| Research sample | This study was a meta-analysis it synthesised effect sizes from multiple samples. Given we extracted information from 281 samples, it would be impractical to provide full details here. We provide information about individual samples included in the meta-analysis in Supplementary Table ST5 |
| Sampling strategy | We conducted a systematic search of the following electronic databases: Web of Science, Scopus, PsycInfo, Business Source Ultimate, Proquest Dissertations and Theses Global, Medline. Studies were eligible if they measured goal adjustment in any form (e.g., self-reported adjustment capacity, objectively assessed behavioural adjustment) and either one or all of the following: (i) antecedents predicting goal adjustment; (ii) goal-related outcomes; (iii) functional outcomes; (iv) wellbeing outcomes. |
| Data collection | We extracted effect sizes (r) from articles that met our inclusion criteria and synthesised pooled effect size estimates using three-level meta-regression. |
| Timing | The search was initially conducted on May 25, 2022, updated on July 15, 2023 and updated again on May 22, 2025 |
| Data exclusions | no data were excluded |
| Non-participation | N/A |
| Randomization | N/A |

# Reporting for specific materials, systems and methods

We require information from authors about some types of materials, experimental systems and methods used in many studies. Here, indicate whether each material, system or method listed is relevant to your study. If you are not sure if a list item applies to your research, read the appropriate section before selecting a response.

## Materials & experimental systems

| n/a | Involved in the study |
|-----|----------------------|
| ☒ ☐ | Antibodies |
| ☒ ☐ | Eukaryotic cell lines |
| ☒ ☐ | Palaeontology and archaeology |
| ☒ ☐ | Animals and other organisms |
| ☒ ☐ | Clinical data |
| ☒ ☐ | Dual use research of concern |
| ☒ ☐ | Plants |

## Methods

| n/a | Involved in the study |
|-----|----------------------|
| ☒ ☐ | ChIP-seq |
| ☒ ☐ | Flow cytometry |
| ☒ ☐ | MRI-based neuroimaging |

## Plants

Seed stocks — *Report on the source of all seed stocks or other plant material used. If applicable, state the seed stock centre and catalogue number. If plant specimens were collected from the field, describe the collection location, date and sampling procedures.*

Novel plant genotypes — *Describe the methods by which all novel plant genotypes were produced. This includes those generated by transgenic approaches, gene editing, chemical/radiation-based mutagenesis and hybridization. For transgenic lines, describe the transformation method, the number of independent lines analyzed and the generation upon which experiments were performed. For gene-edited lines, describe the editor used, the endogenous sequence targeted for editing, the targeting guide RNA sequence (if applicable) and how the editor was applied.*

Authentication — *Describe any authentication procedures for each seed stock used or novel genotype generated. Describe any experiments used to assess the effect of a mutation and, where applicable, how potential secondary effects (e.g. second site T-DNA insertions, mosiacism, off-target gene editing) were examined.*

