## [Peer Review File · Nature Human Behaviour]

A Meta-Analytic Review and Conceptual Model of the Antecedents and Outcomes of Goal Adjustment in Response to Striving Difficulties

Corresponding Author: Professor Nikos Ntoumanis

Version 0:

Decision Letter:

17th April 2025

Dear Dr Riddell,

Thank you once again for your manuscript, entitled "Integrating Goal Adjustment: A Meta-Analytic Review and New Conceptual Model", and for your patience during the peer review process.

Your Article has now been evaluated by 3 referees. You will see from their comments copied below that, although they find your work of potential interest, they have raised quite substantial concerns. In light of these comments, we cannot accept the manuscript for publication, but would be interested in considering a revised version if you are willing and able to fully address reviewer and editorial concerns.

We hope you will find the referees' comments useful as you decide how to proceed. If you wish to submit a substantially revised manuscript, please bear in mind that we will be reluctant to approach the referees again in the absence of major revisions. We are committed to providing a fair and constructive peer-review process. Do not hesitate to contact us if there are specific requests from the reviewers that you believe are technically impossible or unlikely to yield a meaningful outcome.

To guide the scope of the revisions, the editors discuss the referee reports in detail within the team, including with the chief editor, with a view to (1) identifying key priorities that should be addressed in revision and (2) overruling referee requests that are deemed beyond the scope of the current study. We hope that you will find the prioritised set of referee points to be useful when revising your study. Please do not hesitate to get in touch if you would like to discuss these issues further.

1. Reviewer 1 raises core concerns about your search and inclusion criteria. In line with their feedback, we ask that you update your search to include studies published until now and revisit your exclusion criteria. We ask that you avoid exclusions due to pure convenience and explicitly motivate exclusion criteria.
2. Reviewer 3 raises an important concern about bidirectional causality. To address this concern, please conduct analyses that examine if longitudinal and/or experimental studies provide clear evidence for a bidirectional effect after removing cross-sectional correlational studies from the analysis, as requested by the reviewer.
3. While the reviewers find your work interesting, they find the presentation of your work should be improved. Please revise your text to be concise and ensure that you clearly emphasize the contribution of your work.

If you wish to submit a suitably revised manuscript, we would hope to receive it within 4 months. I would be grateful if you could contact us as soon as possible if you foresee difficulties with meeting this target resubmission date.

- Include a "Response to the editors and reviewers" document detailing, point-by-point, how you addressed each editor and

referee comment. If no action was taken to address a point, you must provide a compelling argument. When formatting this document, please respond to each reviewer comment individually, including the full text of the reviewer comment verbatim followed by your response to the individual point. This response will be used by the editors to evaluate your revision and sent back to the reviewers along with the revised manuscript.

- Highlight all changes made to your manuscript or provide us with a version that tracks changes.

Link Redacted

Thank you for the opportunity to review your work. Please do not hesitate to contact me if you have any questions or would like to discuss the required revisions further.

Sincerely,

[Redacted]

[Redacted]

Nature Human Behaviour

Reviewer expertise:

Reviewer #1: goal behaviour/adjustment – antecedents and outcomes ; systematic reviews and meta-analyses

Reviewer #2: goal behaviour/adjustment – antecedents and outcomes

Reviewer #3: goal behaviour/adjustment – antecedents and outcomes

REVIEWER COMMENTS:

Reviewer #1 (Remarks to the Author):

I thought this was an interesting manuscript with valuable insight into goal adjustment. I had a few issues with methodological choices. Overall I think this is a valuable contribution to the field.

Key results:

- A conceptual model of goal adjustment, backed up with meta-analytical evidence, is presented

Validity

- The model has high validity – I have two concerns:

o the exclusion of studies without a strong rationale other than convenience. I know that the majority of research is from Westernised, Educated, Industrialised, Rich and Democratic countries and I accept that review will reflect that. However, to exclude papers not written in English and not available in the author's University subscriptions (which may be some of those lower impact journals that authors from non WEIRD countries might publish in) seems a choice rather than a state of literature that we have to work around.

o The categorisation of the antecedents isn't explained in detail and there is some vagueness in the descriptions of variables and some inconsistencies

Originality & significance

- The paper adds value to the literature by meta-analysing existing published data to produce a conceptual model of goal striving. This tests out our assumptions from other theories. Goal setting spans many different fields beyond psychology so the paper will be of wide interest.

Abstract

- It wasn't clear from this what type of study was included in the review

Introduction

- This is a well argued introduction to the topic that brings together theories of goal adjustment, explains their limitations and the need for this review

o Minor suggestion - Paragraph 1 – a further option, when encountering obstacles, is to abandon their goal completely

Methods

- The method is comprehensive. I have one major issue around the exclusion criteria that may introduce bias make the model less representative

- o What were the justifications for excluding studies not written in English? Excluding these could mean that the theory is yet another one that is not relevant to non English speaking cultures. How many articles were excluded for this reason? I don't think it's enough to just have this as a limitation when it is within your power to include these papers.
- o Excluding articles that weren't available from the university library or directly from the author seems that it potentially could miss too many articles (dependent upon the extent of the University's subscription services). There are other inexpensive ways of locating papers. Again, this produces a bias in the studies included to build the theory.
- I also have a few more minor issues
- o The searches are out of date – could these be updated to incorporate the studies from the last 20 months to bring it up to date
- o Why weren't other methods of detecting relevant papers conducted e.g., ascendancy/ descendancy searches?
- o Could an example and/or definition of a functional outcome (reported as functioning elsewhere) be given where it is first mentioned
- o What was the level of agreement in data extraction and study quality?
- o How as flexible goal adjustment operationalised if measures also included disengagement and reengagement were not separated out
- o For goal re-engagement is this only for goals in the same domain (e.g., adjusting a goal from exercising 3x per week to 2x per week; or adjusting the goal from running to walking 3x per week), or could this be a different goal with the same overarching outcome (e.g., adjusting a goal to exercise 3x per week to eating 1200 per day for the outcome of weightloss) or could it be a complete reevaluation of priorities (e.g., disengaging from the goal to exercise 3x per week to see friends 3x per week)?
- o For the studies that measured goal re-engagement were the goal attainment outcomes for the original or the new goal?
- o Quality rating: what kind of quality aspects does the tool measure. Are their thresholds for poor, moderate and high quality?
- o Does country of origin relate to nationality of population sampled? Or is this where the authors are based?

Results

- The results are well articulated. I have one major issue around the grouping of the antecedents as there seems some inconsistencies. What was the process of determining these? How were disagreements solved? How can the inconsistencies be explained?
- o Agency inhibiting beliefs includes Luck, personal agency belief-luck and personal agency – luck. I wondered why they had been separated this way rather than all categorised as Luck
- o Some of these are quite vague e.g., what does belief system mean; e.g., what kind of locus of control?
- o The affective antecedents aren't included. I understand how they are categorised but it would be useful to have the full list to see what has been explored in the literature
- o Why is SES agency and not a contextual barrier?
- o Why is type D personality in agency and not in personality?
- o Are these categorised always as high = more inhibition / enhancement? E.g., SES positively correlated with positive outcomes; e.g., high religiosity correlated with negative outcomes?
- Could the strength of associations be added to Figure 2
- Could the overall category statistics be added into the text

Discussion

- The discussion is a good attempt to explain the model and seemingly incongruous results. The conclusions are a good reflection of the data.

Other

- There is the odd spelling mistake /typo in the manuscript (e.g., moder instead of moderate)

Reviewer #2 (Remarks to the Author):

I had reviewed a previous version of the manuscript submitted to another journal. The authors had made extensive changes, and I think that the new manuscript, submitted to Nature Human Behaviour did a great job in presenting the results of this extremely important endeavor. The meta-analysis is extensive and informative. It provides a better understanding of the antecedents and consequences of goal engagement, disengagement and flexibility, and highlights promising avenues and research gaps. I especially appreciated how the idea of goal levels was introduced and addressed in the discussion (as a way to make sense of some of the findings). I expect this paper will thus be very valuable to the field. I had a few mostly smallish comments that I hope can help you further improve this manuscript.

-it would be helpful to note early on (perhaps on p.4-5 when you describe different forms of goal adjustment) that disengagement and reengagement in this tradition (especially in the research by Wrosch & colleagues, where it is assessed as an individual difference) typically refers to general tendencies and is not about engagement with or disengagement from a specific goal. This has implications for contextualizing the existing research (so could be returned to in the discussion). There is a difference in what someone does for a specific goal vs. the person's general tendencies; unfortunately the current research does not allow you to distinguish these (since most research focuses on tendencies), and this could be emphasized further in the introduction and discussion sections of the manuscript.

-I appreciate how you grouped the different variables into categories (rather than examine separate specific variables with few studies in each), and agree that this is the better approach. I would have liked, however, a separate table in manuscript that names every variable that went into each category (and how many studies it was in). Otherwise some of these

categories seem pretty nebulous.

-I think that too much attention is drawn in the text to potential moderators, especially since some of the results by moderator may instead be due to the specific variable examined. For example, if studies from Germany were more likely to include action crises (as opposed to another precursor variable), the moderation analysis would show stronger effects from Germany, where the strength of the effects is due to the specific variable examined. Having the information in the text also makes it a bit long, reducing the focus on the more important results. I suggest moving it to the tables instead (perhaps another column in tables 1 & 2 stating significant moderators of the effects).

-You present on p. 21 a nice discussion of your evaluation of publication bias. I wondered how problematic/impactful that bias may be? There are many different bias corrections that attempt to estimate effects corrected for bias, (some much more stringent/biased than others), so I am reluctant to recommend that you include some sort of corrected estimate in the table, but maybe a likely range of effects corrected for bias? Or explain why you're not doing it.

Reviewer #3 (Remarks to the Author):

This manuscript presents a meta-analysis of both the antecedents and consequences of 3 forms of goal adjustment – goal disengagement, goal reengagement, and flexibility. It presents a novel conceptual organization of antecedent variables, which helps to create a tentative model of how different antecedent variables are related to different forms of goal adjustment. It also extends beyond prior meta-analyses (Cheng et al., 2014; Barlow et al., 2014) that focused on the implications of goal-adjustment for wellbeing outcomes, and also considers implications for Goal Attainment and “Functioning” – which appears to refer to physical health, social relationships, and work-related outcomes.

This manuscript got off to somewhat of a slow start, as I was initially unsure what it contributed beyond prior meta-analyses. But by the end of the introduction, the authors finally told me they were organizing research on the antecedents of goal-adjustment, and I was quickly pulled in and even enthralled by later portions of the paper. I spent far more time than I planned on this review, but actually enjoyed the experience greatly. Broadly, then, some version of this manuscript should be published, but there are some substantial issues that the authors will need to successfully attend to before this should be done.

First, the authors need to overcome their “slow start”. For starters, the title and the abstract don't tell the reader what people are “adjusting” their goals in response to (i.e., obstacles and difficulties achieving one's goals), which would create a great deal of confusion about what the paper is even about for many potential readers. For me, this was an easy issue to overcome because I am familiar with this literature. But many potential readers won't be, and this will hamper the article's ability to impact the literature.

Second, neither the abstract nor the opening sections of the introduction clarify the unique contribution of the current meta-analysis beyond prior meta-analyses. This should be front and center from the very beginning. They organized potential antecedent variables, expanded the outcome variables under consideration to also include goal-progress and functioning, and used the results to create an integrative model. No prior meta-analysis has done that. Don't make me search until the end of the introduction to figure this out.

Relatedly, the authors should emphasize the relative dearth of theorizing on antecedents of goal-adjustment much earlier and more clearly. This is the main reason why this paper has the potential to contribute greatly to the literature.

Beyond that, I would encourage the authors to break apart the “Functioning” and “Impairment” outcome variables more, to whatever extent is possible with the studies available for this meta-analysis. As I noted above, this appears to refer to a heterogeneous collection of variables, including physical health, social relationships, work-related outcomes, and possibly others (e.g., academic outcomes, I would venture to guess). These are of great interest to researchers in different fields – e.g., health psychology, relationship science, industrial/organization psychology; not to mention other fields entirely like behavioral health and management. While researchers in these fields might be truly enthralled to hear what this meta-analysis can tell them relevant to their outcome variables of interest, they might see very little value in the variable of “functioning”.

More modest issues with introduction:

Haase et al. (2013) reviewed and integrated an overlapping set of theories and arrived at 3 very similar constructs to Scobbie et al. (2021) (which the authors use to organize research on different forms of goal-adjustment). This should be cited and acknowledged. Currently, this seems to be cited as an extension of Brandtstadter's model, but it is an integration with other models.

Page 5-6: The authors say that changes in goal-striving are the mechanism that links goal adjustment to wellbeing and cite Carver & Schier (2005), but I don't follow. Disengagement and reengagement are both seen as beneficial for wellbeing, but they should (theoretically speaking) affect goal-striving in opposite directions, correct? The example previously prior to this sentence seem to involve interactions between goal-disengagement/reengagement and goal-achievement/failure in

predicting wellbeing, which the authors don't consider in this meta-analysis.

Methods:

I'm not a methodological expert in systematic review or meta-analysis, but have of course read enough of them over my career. The authors' procedures seemed largely appropriate, but I encourage the editor to gather reviews from someone with greater methodological expertise here. I would defer to their opinion on these issues.

Question of Causal Direction: Affect is considered as both a possible antecedent of goal-adjustment (Affect-Enhancing, Affective-Diminishing) and as a possible outcome of goal-adjustment (Wellbeing, Illbeing). The same is also true of Goal Progress – i.e., Goal-Enhancing/Inhibiting antecedents (and possibly Agency?) vs. Goal-Attainment/Failure outcomes. When a study included in this meta-analysis examined these variables experimentally or longitudinally, it is quite straightforward to code specific effects for inclusion into one analysis or the other. But the current authors also note that they included cross-sectional correlational studies, and that they seem to have used the original papers' authors hypotheses to sort effects into the antecedent vs. outcome meta-analyses. First, is that an accurate description of the current procedures? If so, this seems potentially problematic, as the authors of the original articles could of course be wrong and the effects may pertain to the opposite direction of causality in reality. This could spuriously increase effects that should only be listed in one direction of causality (or perhaps neither), but are currently listed in both directions. Looking at Figure 2, Affect-Enhancing antecedents are related to both Reengagement and Flexibility, and, likewise, Wellbeing consequences are related to both Reengagement and Flexibility. Likewise, Goal-Enhancing antecedents, Agency-Enhancing antecedents, and Goal-Attainment outcomes are all related to Goal Reengagement. I'd like to see the authors conduct/report analyses (in the paper itself, not in the supplement) that examine if longitudinal and/or experimental studies provide clear evidence for a bidirectional effect or if these effects are actually unidirectional (or even consistently non-significant) after removing cross-sectional correlational studies from the analysis.

Results and Discussion:

The organization of different antecedent variables in the "Conceptual Model" section is quite useful, as is Figure 2!

Tables: Why are the variables in Table 1 (and Table 2) listed in different orders for different goal-adjustment variables? Significant effects appear to be aggregated at the top of each panel (except for Affect Enhancing-Flexibility relationship in Table 1, oddly), but they're not in order of effect size or alphabetical otherwise. I suggest a more systematic approach, so readers looking for a particular variable can find it more easily. Perhaps place the antecedent/outcome variables in alphabetical order, and highlight significant effects so that they "pop out" for readers who are just interested in finding significant effects (e.g., star the significant effects or put them in bold font).

It would be useful if the Results section text told the reader in which section of the Supplementary Information they can find specific moderator analyses – e.g., the Antecedents of Goal Disengagement results section could tell me where to find the effects for other goal-inhibiting variables besides Action Crisis (if there were any available), and the methodological moderation analysis (comparing lab, longitudinal, experience-sampling studies). I was unable to find or locate the more detailed presentation of the moderation analyses.

P16: Antecedents of Goal Reengagement section: Optimism is listed as an affect-enhancing variable here, but the "Conceptual Model" section above lists it as an adaptive personality trait. Is there perhaps a distinction being made here between Dispositional/Trait Optimism and Goal-Specific Optimism? If so, more precise terminology should be adopted to clarify this. Or is this just a mistake?

It is odd that "Goal-Inhibiting" antecedent variables are associated with greater Goal Reengagement. More explanation is required, in terms of the variables that constitute this antecedent category. This is somewhat complicated by the fact that no individual Goal-Inhibiting variable alone is linked with Goal-Reengagement, but more information is needed nonetheless. For example, I could imagine that an Action Crisis with Goal #1 could lead to Goal-Reengagement with Goal #2 (presumably after one also disengaged from Goal #1). Do all (or even most) of the Goal-Inhibiting variables refer to the previous goal (i.e., Goal #1) and not to the Reengaged goal (i.e., Goal #2)?

The authors attempt to address this issue in the Discussion section (on page 25 of the pdf). I honestly had some difficulty following this discussion section, so, first, I encourage the authors to use examples to make their discussion clearer. For example, they state: "Goal reengagement encompasses both the continuation of goal striving at a high level of one's goal hierarchy and the discontinuation and subsequent adjustment of striving at lower levels." To flesh this out, perhaps a concrete example would be useful – e.g., perhaps a person still wishes to exercise and improve their physical health. An injury prevents them from running anymore, so they switch to swimming? (Not entirely sure if that corresponds to their intended, abstract description.) The next sentence reads, "Viewed in this light, it follows that both processes involved in the cessation of old modes of goal striving and processes that reenforce striving for the goal at a higher conceptual level contribute to goal reengagement." If I'm following the authors correctly, does this mean that they hypothesize that "Goal-Enhancing" antecedent variables related to the Higher-Order Exercising goal (in the previous example) would be related to Reengagement (but not the "Goal-Enhancing" variables related to the Lower-Order Running goal). Likewise, does it mean that they hypothesize "Goal-Inhibiting" variables related to the Lower-order Running goal are related to Reengagement (but not "Goal-Inhibiting" variables related to the Higher-Order Exercise goal)? If this is the case, I again wonder if the authors can examine the actual studies in this meta-analysis to determine if this hypothesis is supported. A formal moderation meta-analysis would be preferable if possible, of course, but even more informal observations would be useful here. I also wonder if the authors will need to consider whether Goal-Inhibiting variables are related to two different Lower-Order goals (e.g., running vs. swimming in my prior example; or the previous and the new lower-order goal more broadly), and if these two

lower-order goals are always necessarily related to the same higher-order goal at all?

Regarding the surprising null relationship between Goal-Reengagement and Goal Progress Outcomes: On page 25-26 of the discussion, the authors note (similar to the previous point) that this could be due different studies examining goal progress for different goals (i.e., the original vs. reengaged goal) at different times (before vs. after reengagement). I again wonder if the authors can code goal-progress outcomes to shed some light on this issue. Perhaps some of the goal-progress measures are too non-specific to be coded in this manner, but if there are a sufficient number that can be coded in this manner, it would be extremely useful.

The effects of Goal-Disengagement on "Impairment" need to be broken apart in the main results section more. As I mentioned in relation to the introduction and the framing of the paper more broadly, Health psychologists, Relationship scientists, and I/O psychologists will want to know if these effects pertain to their domain of interest.

Moreover, this effect is conceptually important to break apart and understand further. The effect of Goal Disengagement on Goal-Progress is basically tautological by itself. But understanding the effects of Disengagement on Impairment goes beyond mere goal progress and could highlight ways in which Disengagement truly is a very mixed bag. Sure, it might lower Depression and Anxiety (Illbeing), but these results seem to imply it MIGHT simultaneously reduce one's lifespan, result in getting fired from one's job, and/or lead to a divorce, perhaps???? If this is the case, Goal-Disengagement appears more similar to some societal stereotypes of the indifferent slacker – i.e., who is unemployed, single, content to live in his parents' basement till age 40, eating too many Cheetos, playing video games nonstop, etc.; rather than the more positive portrait that Wrosch and others paint. I was especially surprised to see that, later in the article, the authors did not discuss the relationship between Disengagement and Impairment in the relevant discussion section.

More broadly, the individual outcome variables constituting each outcome category should be consistently broken apart (whenever possible), as the authors did for the antecedent variable categories. This could also shine further light on the Flexibility-Functioning relationships, for example.

Finally, the Supplemental Table (with file name ending in 262818) listing the specific antecedent variables in each category was useful, but it appears to be missing Affect Enhancing and Affect Diminishing variables. I also happened to notice that "Type D Personality" is listed as an Adaptive Personality variable, but also an Agency Enhancing variable. (Which I only noticed because I have never heard of Type D personality before. I've heard of Type A and Type B, but not D or even C.) Also, Optimism is not listed as an Adaptive Personality variable, but is in the text. Regardless, the important points are – are the authors actually counting variables in two or more categories? This seems inappropriate, as it would muddy the literature rather than clarify it. Or are there just simply mistakes in various places that need to be corrected (hopefully only in the paper itself, but if there are mistakes in coding of the meta-analysis itself, some aspects of it would of course need to be re-done)?

Version 1:

Decision Letter:

Our ref: NATHUMBEHAV-25031157A

25th July 2025

Dear Dr. Riddell,

Thank you for submitting your revised manuscript "A Meta-Analytic Review and Conceptual Model of Goal Adjustment in Response to Striving Difficulties" (NATHUMBEHAV-25031157A). It has now been seen by the original referees and their comments are below. As you can see, the reviewers find that the paper has improved in revision. We will therefore be happy in principle to publish it in Nature Human Behaviour, pending minor revisions to satisfy the referees' final requests and to comply with our editorial and formatting guidelines.

We are now performing detailed checks on your paper and will send you a checklist detailing our editorial and formatting requirements within two weeks. Please do not upload the final materials and make any revisions until you receive this additional information from us.

Sincerely,

Reviewer #1 (Remarks to the Author):

Thank you for your careful consideration of my comments. You have addressed these in the review

Reviewer #2 (Remarks to the Author):

The authors did an excellent job addressing all my comments. I have one more small lingering observation: In the definitions of the key constructs (engagement, disengagement, flexibility), both engagement and disengagement are defined as processes (p. 3, p. 5), while flexibility is defined as “an individuals’ general disposition” (p. 3 & 4) – it would be helpful if the authors explicitly acknowledged this discrepancy and discussed potential implications.

Reviewer #3 (Remarks to the Author):

I was previously listed as Reviewer #3. This revision is very responsive to many of my previous concerns, so I thank the authors for their efforts and diligence in this! The clarification of the purpose of the meta-analysis and conceptual model earlier on is helpful, as is the addition of sensitivity analyses focused on experimental/longitudinal studies to better establish the order of causality. I appreciated the authors’ point that many of the questions I posed to them are ultimately issues in the broader literature – authors need to do a better job of specifying whether disengagement/reengagement/goal-progress/etc. is linked to an original goal or an alternative goal. This should help to improve research practices in this area moving forward (hopefully, of course).

My only remaining feedback is that I thought the authors could “go further” in addressing some of my original concerns. In most cases, this is minor, so I will leave it to the editor’s discretion (of course) whether to require this or not. In all cases, the goal is clear presentation to help the reader better understand the paper, rather than a substantive concern with the underlying science.

-The opening paragraphs now do an excellent job of highlighting the novel purpose of this paper. The revised title and abstract are certainly improved, but they are still missing some of the critical emphasis that comes across in the opening paragraphs. I realize it’s difficult to fit everything in a short title and short abstract, but I would encourage the authors to further revise these to better highlight the novel contribution – i.e., the fact that they are presenting a novel conceptual model of antecedent variables and extending the outcomes considered beyond wellbeing to also include goal-progress and functioning.

-In their response letter, the authors indicate that “We reorganised the tables so that effects are in order of effect size and significant effects are highlighted”. This is certainly true for Table 1, but it does not appear to be completely true for Table 2. That is, significant effects are highlighted in Table 2, but outcome are now listed in a standardized order for Table 2. I encourage the authors to use a standard system across tables to help the reader find variables of interest to them.

-Finally, I vaguely recall that it was easier to figure out what common measures of goal engagement, disengagement, and flexible striving were for the included studies, in the prior version of the paper (so I did not note this as an issue). But I had more trouble finding out what they were in this version. Are most studies (that used a standardized, trait-like measure of these tendencies) based on Wrosch et al. (2003) and Brandtstadter & Renner (1990)’s scales, for example?

Reviewer points for improvement of the MS are highlighted in red. Our responses to these comments are provided in blue. Changes have been tracked in the manuscript and additions are highlighted in light blue.

Reviewer #1 (Remarks to the Author):

I thought this was an interesting manuscript with valuable insight into goal adjustment. I had a few issues with methodological choices. Overall, I think this is a valuable contribution to the field.

RESPONSE:

We are grateful for your overall positive evaluation of our work, and your detailed and valuable feedback. It is with great difficulty that we refrained from thanking you profusely after each comment you made. In the revision, we did our best to address your misgivings.

Key results:

- A conceptual model of goal adjustment, backed up with meta-analytical evidence, is presented

Validity

- The model has high validity – I have two concerns:

o the exclusion of studies without a strong rationale other than convenience. I know that the majority of research is from Westernised, Educated, Industrialised, Rich and Democratic countries and I accept that review will reflect that. However, to exclude papers not written in English and not available in the author's University subscriptions (which may be some of those lower impact journals that authors from non WEIRD countries might publish in) seems a choice rather than a state of literature that we have to work around.

RESPONSE:

We appreciate the reviewer's concern regarding the exclusion of potentially relevant literature from non-WEIRD contexts and lower-impact journals. In response, we re-examined all studies retrieved through our search strategy and included two non-English language articles that met our inclusion criteria. As we write pages 9-10:

“A further methodological deviation in the present review was the inclusion of articles not written in English. We made this decision to minimise potential publication language biases and enhance the comprehensiveness of the review. The authorship team is proficient in English, Danish, French, German, Greek, Hindi, and Tamil. We translated one article published in Japanese, a language not spoken by any team member, using a large language model (ChatGPT, version 4o).”

We also clarified that, where full-text access was unavailable from our university libraries, we used the institutional document request service to obtain the relevant articles. This service

ensures a comprehensive search beyond convenience or accessibility constraints. In addition, we checked ResearchGate for full text of articles (pg. 10). In all, we made a robust and transparent effort to include all eligible studies within the scope of our review without introducing systematic exclusion based on language or journal impact.

o The categorisation of the antecedents isn't explained in detail and there is some vagueness in the descriptions of variables and some inconsistencies

RESPONSE:

We acknowledge that our description of the categorisation process for antecedents was brief in the previous submission. We expanded the description in the revision under *Conceptual Model Development and Data Analysis* (pg. 13-14):

“We developed our integrative model of goal adjustment both inductively, drawing the data collected in our literature search, and deductively, based on meta-analytic associations identified in the literature. To develop our model, the first and last authors, both of whom are experts on goal adjustment, first reviewed all extracted antecedent variables and developed thematic categories that aligned with well-established but broad psychological constructs (e.g., personality, affect, agency), processes (e.g., goal striving), and contexts (e.g., development/environmental context). The authors then collaboratively assigned individual antecedent variables into these categories. The authors further divided these antecedents based on whether primary studies and the broader psychological literature views them as enhancing/adaptive or inhibitory/maladaptive, culminating in the following categories: goal enhancing/inhibiting variables; affect enhancing/diminishing variables; adaptive/maladaptive personality traits; developmental and contextual barriers/enablers; agency enhancing/inhibiting variables. The categorisation of variables was refined through an iterative process that involved consulting primary studies and examining the theoretical frameworks used by original authors to contextualise their variables. Final categorisation decisions were reviewed and approved by all manuscript authors. We similarly grouped outcomes into six overarching categories, conceptualised as three opposing pairs: wellbeing versus illbeing, functioning versus impairment, and goal progression versus goal reduction. We present a detail mapping of individual variables to model constructs in Supplementary Information (ST-1), and a summary of the variables assessed across primary studies in Supplementary Information (ST-2).”

We spotted an inconsistency in the table presenting the antecedent categorisations. This was due to an oversight whereby we included an old, incomplete version of the table. Importantly, this table did not reflect the coding we used in the actual datasheet; that is, the analysis used the correct antecedent classifications. We updated the table to remove inconsistencies and checked all other tables for accuracy.

Originality & significance

- The paper adds value to the literature by meta-analysing existing published data to produce a conceptual model of goal striving. This tests out our assumptions from other theories. Goal

setting spans many different fields beyond psychology so the paper will be of wide interest.

RESPONSE:

We thank you for your positive evaluation of the originality and significance of our work.

Abstract

- It wasn't clear from this what type of study was included in the review

RESPONSE:

We included experimental, cross-sectional, and longitudinal quantitative studies that examined any antecedents or outcomes of goal adjustment. We added this information to the Abstract.

Introduction

- This is a well argued introduction to the topic that brings together theories of goal adjustment, explains their limitations and the need for this review

o Minor suggestion - Paragraph 1 – a further option, when encountering obstacles, is to abandon their goal completely

RESPONSE:

In agreement, we added this sentence to the Introduction (pg. 2):

“In response to these difficulties, people may persist in their efforts, or they may adjust their goals—modifying, reprioritising, or abandoning them altogether.”

Methods

- The method is comprehensive. I have one major issue around the exclusion criteria that may introduce bias make the model less representative

o What were the justifications for excluding studies not written in English? Excluding these could mean that the theory is yet another one that is not relevant to non English speaking cultures. How many articles were excluded for this reason? I don't think it's enough to just have this as a limitation when it is within your power to include these papers.

o Excluding articles that weren't available from the university library or directly from the author seems that it potentially could miss too many articles (dependent upon the extent of the University's subscription services). There are other inexpensive ways of locating papers. Again, this produces a bias in the studies included to build the theory.

RESPONSE:

Please see our response to your first comment.

- I also have a few more minor issues

- o The searches are out of date – could these be updated to incorporate the studies from the last 20 months to bring it up to date

RESPONSE:

We agree with the reviewer and so updated the search. It identified a further 25 articles published in the last ~2 years that contributed an additional 210 effects to our meta-analysis. This addition strengthened the manuscript.

- o Why weren't other methods of detecting relevant papers conducted e.g., ascendancy/descendancy searches?

RESPONSE:

We thank the reviewer for pointing out this omission. We did conduct ascendancy/descendancy searches, as per our preregistered protocol, and we now mention it (pg. 10):

“We supplemented our main search with an ascendancy/ descendancy search, assisted by the Paperfetcher web app (Pallath & Zhang, 2023).”

- o Could an example and/or definition of a functional outcome (reported as functioning elsewhere) be given where it is first mentioned

RESPONSE:

In our preregistered protocol, we defined functional outcomes as follows:

“Functional Outcomes include all outcomes that relate to the physical, social, or societal functioning of the individual that follow from goal adjustment. Functional outcomes related to physical health (including pain) are included in this definition. Example functional outcomes could include job performance, physical health indicators, and social integration.”

We intended the function/impairment to reflect biological and social indicators of health and livelihood, and the function wellbeing/illbeing to reflect psychological indicators of health and livelihood. We differentiated these categories, because the literature on goal adjustment often treats physical and psychological outcomes separately. For example, models on the evolution of goal adjustment over the lifetime, such as Brandstadter and Renner's (1990) dual process model or Shane and Heckhausen's (2019) model of lifespan development, posit that goal striving flexibility may be needed to adapt to physical limitations associated with aging, and that, if executed effectively, this accommodation will promote psychological wellbeing. We expanded upon our definition of functional outcomes as follows (pg. 16-17):

“The outcome variables we used are largely self-explanatory. Goal progression and goal reduction refer to variables capturing the effective versus ineffective pursuit or maintenance of personal goals (e.g., goal progress vs. self-handicapping). Wellbeing and illbeing encompass psychological indicators of mental health and emotional welfare, such as life satisfaction and depressive symptoms, respectively). In contrast, functioning and impairment reflect biological and social dimensions of health, including variables that describe individuals’ physical capabilities and social integration (e.g., experiencing burnout and forming supportive). We distinguished mental wellbeing/illbeing from physical and social functioning/impairment based on theoretical perspectives that posit unique pathways linking goal adjustment to different domains of functioning. For example, both the dual process model of self-regulation and lifespan development theory suggest that flexible goal adjustment plays a central role in adapting to age-related declines in physical capacity (Brandtstädter & Renner, 1990; Shane & Heckhausen, 2019).”

o What was the level of agreement in data extraction and study quality?

RESPONSE:

Unfortunately, we did not track this information and so could not add it to the manuscript. We noted this issue as a limitation but clarified that all individuals involved in data extraction and study quality coding were experienced researchers who had authored meta-analyses and engaged in prior data extraction and study quality coding. Thus, although there were minor discrepancies in data extraction and study quality, we were easily able to resolve them via discussion (pg. 11).

o How as flexible goal adjustment operationalised if measures also included disengagement and reengagement were not separated out

RESPONSE:

We used the concept of “flexible goal adjustment” to capture combined, overarching, or multifaceted processes of goal modification, which cover the broader suite of processes that individuals implement to “get around” (rather than through) obstacles (Brandtstädter & Renner, 1990). We differentiate flexible goal adjustment from goal disengagement and reengagement, which are more bounded around distinct processes. Where we could identify distinct disengagement and reengagement sub-processes (e.g., the disengagement and reengagement subscales of the Goal Adjustment Scale), we coded them as disengagement and reengagement separately. Where disengagement and reengagement were combined (e.g., the Flexible Goal Adjustment subscale of the Ten-Flex scale includes items reminiscent of both reengagement and disengagement such as “When things don’t go as planned, I quickly find new ways to move forward” and “I can let go of goals that prove to be unattainable”, that are not typically dissociated from one another), we coded them as flexible goal adjustment. We clarified as follows (pg. 12):

We defined flexible goal adjustment as an individual’s general propensity toward flexible adjustment of goal striving or accommodative goal striving. Studies that measured his construct did not disentangle separate processes of goal adjustment (e.g.,

disengagement, reengagement). Where we could identify distinct disengagement and reengagement sub-processes, we coded them separately as disengagement and reengagement. Where disengagement and reengagement were combined or were not clearly separable within scale items, we coded them as flexible goal adjustment.

o For goal re-engagement is this only for goals in the same domain (e.g., adjusting a goal from exercising 3x per week to 2x per week; or adjusting the goal from running to walking 3x per week), or could this be a different goal with the same overarching outcome (e.g., adjusting a goal to exercise 3x per week to eating 1200 per day for the outcome of weightloss) or could it be a complete reevaluation of priorities (e.g., disengaging from the goal to exercise 3x per week to see friends 3x per week)?

RESPONSE:

The reviewer has identified an important but overlooked issue: the line between disengagement and reengagement is not as clear-cut as one would like it to be, and dis/reengagement can take many forms. A common popular culture example involves an individual who decides to give up on the goal of finding a romantic partner and instead to focus on domains of self-improvement, such as career or health. Has this individual disengaged from their original goal and reengaged in something else? Or will they eventually resume pursuit of a partner once they feel they have progressed on the self-improvement goals? Capturing such phenomena, some authors have provided evidence for goal shelving and freezing (Davydenko et al., 2019; Mayer & Freund 2022), where goals are temporary put aside to meet other demands or opportunities.

What authors examine when they investigate goal disengagement and particularly reengagement is often ambiguous. We cannot fully address the reviewer's question, in part because sometimes we were also unsure as to whether primary study authors measured reengagement within the same goal domain or a different one. It appears that researchers would need to specify "disengagement from what" and "reengagement with what."

Acknowledging that reengagement can take many forms, we broadly defined it as "*behaviorally resuming pursuit with a new or modified goal*", which would cover both of the situations the reviewer provided in their example. We did so, in part, to align with the position put forth by key researchers in the reengagement literature, such as Carver & Scheier and Wrosch. We touched further on the nuances around goal reengagement and considered emerging concepts such as goal shelving and goal freezing (pg. 31):

"Our findings regarding goal facilitatory and inhibitory antecedents indicate that the relation between antecedents and disengagement or reengagement depends on whether goal striving is examined at the micro or macro level. This implies that a multilevel approach to studying goal striving may be essential (Carver & Scheier, 1982). A key limitation in the current literature is the lack of clarity regarding the level of the goal hierarchy at which disengagement and reengagement are assessed. Adjustment processes may differ markedly depending on whether they occur within the same hierarchical level (e.g., shifting from regular running to walking) or across levels (e.g., from exercising to dieting to achieve weight loss). These distinctions are consequential, as disengagement may hinder progress at one level, whereas

reengagement can preserve or redirect progress toward higher-order goals. For example, experiments that manipulate goal attainability have demonstrated that individuals often make greater progress towards overarching aims when they relinquish unattainable lower-order goals (Choi et al., 2020; Ntoumanis et al., 2014; Riddell et al., 2022). Moreover, the boundaries between goal progress, disengagement, and reengagement are often more fluid than typically presented. An individual may, for instance, appear to abandon the goal of finding a romantic partner and redirect effort toward self-improvement, yet may not have fully disengaged from the original goal even after months or years of nonpursuit. This conceptual ambiguity is reflected in emerging evidence on goal shelving and freezing phenomena (Davydenko et al., 2019; Mayer & Freund 2025). We contend that a more nuanced understanding of goal adjustment requires researchers to begin by precisely specifying the referent of disengagement and reengagement, namely, disengagement or reengagement with what?

o For the studies that measured goal re-engagement were the goal attainment outcomes for the original or the new goal?

RESPONSE:

Again, the reviewer has identified what we also see as a critical issue in the goal adjustment literature. Often it is not clear whether participants are reengaging with new goals or modified versions of the old goal; it is possible that this is why we found associations between reengagement and both positive and negative goal outcomes. If authors were measuring (or participants were reflecting on) progress of the old goal, we would expect a negative correlation with reengagement; conversely, if authors were measuring progress with a new goal, we should expect a positive association with reengagement. We raised these issues in the Discussion section (see our response above).

o Quality rating: what kind of quality aspects does the tool measure. Are their thresholds for poor, moderate and high quality?

RESPONSE:

Although the authors of this tool did not identify cutoffs for high, moderate, and low quality studies, they did suggest relevant conservative and liberal thresholds. We adapted these thresholds as cut-off points to identify low, moderate, and high quality studies in our review, but noted that we did not exclude studies based on their quality ratings (pg. 12).

“The authors of QualSyst identified conservative ($<.75$) and liberal ($<.55$) thresholds of minimum quality ratings for study inclusion in reviews. Although we did not exclude studies based on quality ratings, we adapted these thresholds as cutoff points for identifying high ($\geq.75$), moderate ($.74 - .56$), and low ($\leq.55$) quality studies.”

o Does country of origin relate to nationality of population sampled? Or is this where the authors are based?

RESPONSE:

The country of origin relates to the location that the sample was taken from (e.g., “German” would refer to studies conducted with individuals living in Germany at the time of the study; presumably participants would be mostly German nationals). We have clarified this in the manuscript (see pg. 15).

“Where possible, we examined the influence the potential moderating influence of several study characteristics: average sample age, proportion of female participants, whether the sample was drawn from a specialised (e.g., clinical, student, or professional) population, the country of origin (i.e., geographic region of the sample), and study quality ratings.”

Results

- The results are well articulated. I have one major issue around the grouping of the antecedents as there seems some inconsistencies. What was the process of determining these? How were disagreements solved? How can the inconsistencies be explained?

RESPONSE:

The Supplementary Information table containing the categorisations belonged to an earlier (i.e., pre-submission) draft and, as the reviewer noted, there were some inconsistencies. Thankfully, these inconsistencies were not mirrored in the main data file; that is, the analysis was based on our finalised categorisation scheme. We corrected the table (see ST-4).

o Agency inhibiting beliefs includes Luck, personal agency belief-luck and personal agency – luck. I wondered why they had been separated this way rather than all categorised as Luck

RESPONSE:

Our general approach was to extract data as they had been reported in the primary studies. That is, where the original authors differentiated sub-categories of an overarching concept, we chose to also extract these sub-categories independently rather than amalgamate them into a single entity/category. We made this decision because we acknowledge that our expertise does not span the multitude of concepts and phenomena uncovered by this review and we trust that, where authors had separated out sub-constructs, there were subject-matter-specific theoretical or empirical reasons to do so. For example, it would be erroneous to group autonomous and controlled goal motivation into the overarching category of “motivation” as there is evidence that they predict goal striving in distinct ways (Sezer et al., 2024).

The aim of our review is to provide broad, encompassing categories that group various types of antecedent variables together to be able to hypothesise about how different facets of the person and the environment contribute to goal adjustment. A similar approach has been taken in other fields of psychology (Albarracín et al., 2024) and by Barlow et al. (2020) in their meta-analysis of wellbeing outcomes of goal disengagement and reengagement. We added this information (pg. 9):

“... although antecedent variables examined in the literature are distinct, there is a degree of overlap in many concepts, while variables often coalesce under broader categories; for example, motivation is a common umbrella category in psychology that encompasses a range of distinct yet related variables. A similar methodological

approach has been employed in other areas of psychology (Albarracín et al., 2024) and by Barlow et al. (2020) in their meta-analysis of wellbeing outcomes associated with goal disengagement and reengagement.

In our main analysis, we grouped related sub-constructs under the same category. For example, we grouped the luck constructs (which the reviewer cites in their example) under agency, as these constructs refer to one's ability to exert influence over their environment. Occasionally, sub-constructs differed on whether they were classified as positive or negative exemplars of a category. We leveraged the rationale provided by primary studies to guide our categorisation. For instance, manuscripts on goal motives consistently state that autonomous and controlled motives predict goal striving in a divergent manner; as such, we categorised autonomous motives as positive exemplars, and controlled motives as negative exemplars, of goal specific antecedents. We included this information on pg. 14:

“We refined the categorisation of variables through an iterative process that involved consulting primary studies and examining the theoretical frameworks used by original authors to contextualise their variables.”

For the ancillary analysis (ST-2, ST-3), where we examined individual predictors within each category, we deferred to the expertise of the primary study authors and retained their classification of constructs into sub-constructs.

Reference

Sezer, B., Riddell, H., Gucciardi, D. F., Sheldon, K. M., Sedikides, C., Vasconcellos, D., Jackson, B., Thøgersen-Ntoumani, C., & Ntoumanis, N. (2024). Goal motives, approach/avoidance appraisals, psychological needs, and well-being: A systematic review and meta-analysis. *Motivation Science*. Advance online publication. <https://doi.org/10.1037/mot0000366>

o Some of these are quite vague e.g., what does belief system mean; e.g., what kind of locus of control?

RESPONSE:

For pragmatic reasons, we used shorthand summaries of antecedent variables when extracting data, which we report in our supplementary table (ST-4). We understand that the reviewer finds some categorisations vague but we feel that, on the whole, our descriptions provide sufficient detail to give a general overview of the types of variables that were amalgamated into our overarching categories. In the *Included Studies* supplementary table (ST-5) we list the antecedent and outcome variables that we examined in each study. Thus, readers curious about specific antecedents, adjustment processes, or outcomes can refer to the primary studies for further information.

o The affective antecedents aren't included. I understand how they are categorised but it would be useful to have the full list to see what has been explored in the literature

RESPONSE:

We now list affective antecedents in the updated supplementary table (ST-4).

o Why is SES agency and not a contextual barrier?

RESPONSE:

As explained in the previous response, we leveraged theoretical justifications in primary studies to guide our classifications. Some authors argued that the effect of SES on goal adjustment was apparent, because SES influences the resources available to an individual to achieve their goals, thus either enhancing or limiting their agentic capacity. For example, imagine two individuals both living in the same city who have the goal of running a marathon. One has a high SES and can dedicate resources to hiring fitness trainers, taking time off work to practice, and buying top-of-the-line equipment. The other has a low SES and lacks access to these resources. Thus, the high SES individual has a greater capacity to control their goal striving. We acknowledge your point that some variables, such as SES, could fit into multiple categories. We modified as follows (pg. 33):

“A related limitation concerns the subjective nature of the variable categorisation process. Categorisation decisions were reached through consensus across multiple researchers with expertise spanning diverse areas of psychology and were guided by the theoretical framing of variables in the primary literature. The classification of antecedents and outcomes as positive or negative exemplars within overarching categories followed the same approach. Again, these decisions were primarily informed by the theoretical positioning adopted by original study author. Nonetheless, we acknowledge that alternative classifications are plausible, and that some variables may reasonably fit within multiple or differing categories. These possibilities warrant empirical scrutiny.”

o Why is type D personality in agency and not in personality?

RESPONSE:

This was an error in the old table, which we carried over from a pre-submission draft. We made the correction.

o Are these categorised always as high = more inhibition / enhancement? E.g., SES positively correlated with positive outcomes; e.g., high religiosity correlated with negative outcomes?

RESPONSE:

We were unclear about this comment. We believe the reviewer is asking whether we coded variables as falling into the inhibiting or enhancing side of a category.

It was not necessarily the case that variables correlated with more inhibition/enhancement. As stated in the *Conceptual Model Development* section, we followed theoretical arguments given in primary studies when categorising variables as being beneficial or maladaptive for goal striving. For several reasons, variables in the wider literature often followed the pattern of higher variables scores = more inhibition/enhancement. Critically, however, we emphasise that we drew on the original hypotheses of primary study authors, not their results. As such, in cases where authors hypothesised a relation in one direction by found another, we based

our categorisation on the hypothesised direction of the effect. Our meta-analysis, then, tests the theoretical assumptions underlying the research, rather than simply confirming the size of effects in one direction or the other. We amended the manuscript to add further detail about the categorisation process as well as a brief discussion highlighting that our work examines evidence for hypotheses proposed in the goal striving literature (see previous response). We write on pg. 14:

“We developed our integrative model of goal adjustment both inductively, drawing from the data collected in our literature search, and deductively, based on meta-analytic associations identified in the literature. To develop our model, the first and last authors, both of whom are experts on goal adjustment, first reviewed all extracted antecedent variables and developed thematic categories that aligned with well-established but broad psychological constructs (e.g., personality, affect, agency), processes (e.g., goal striving), and contexts (e.g., development/environmental context). The authors then collaboratively assigned individual antecedent variables into these categories. The authors further divided these antecedents based on whether primary studies and the broader psychological literature views them as enhancing/adaptive or inhibitory/maladaptive, culminating in the following categories: goal enhancing/inhibiting variables; affect enhancing/diminishing variables; adaptive/maladaptive personality traits; developmental and contextual barriers/enablers; agency enhancing/inhibiting variables. We refined the categorisation of variables through an iterative process that involved consulting primary studies and examining the theoretical frameworks used by original authors to contextualise their variables. Final categorisation decisions were reviewed and approved by all manuscript authors. We similarly grouped outcomes into six overarching categories, conceptualised as three opposing pairs: wellbeing versus illbeing, functioning versus impairment, and goal progression versus goal reduction. We present a detail mapping of individual variables to model constructs in Supplementary Information (ST-4), and a summary of the variables assessed across primary studies in Supplementary Information (ST-5).”

Alternatively, the reviewer may be asking whether a high score could be something else than inhibiting or enhancing, to which our answer is no. We did not think that a third category was sensible, and we are unsure of what an appropriate third category might represent.

- **Could the strength of associations be added to Figure 2**

RESPONSE:

We already present strength of associations in Tables 1 and 2. Our preference is to leave category statistics in table format so that they can be quickly and easily identified in full (i.e., with additional information such as SE and 95% CIs). In alignment with APA formatting guidelines (<https://apastyle.apa.org/instructional-aids/numbers-statistics-guide.pdf>) and to keep the manuscript as uncluttered as possible, we would rather not repeat the same statistics across multiple places in the manuscript. However, if the reviewer or the editor feel strongly against this, we will reconsider.

- Could the overall category statistics be added into the text

RESPONSE:

We already presented category statistics in Tables 1 and 2. As just mentioned, our preference is to present category statistics in table format.

Discussion

- The discussion is a good attempt to explain the model and seemingly incongruous results. The conclusions are a good reflection of the data.

RESPONSE:

Thank you very much for your positive evaluation of our Discussion.

Other

- There is the odd spelling mistake /typo in the manuscript (e.g., moder instead of moderate)

RESPONSE:

We have proofread the manuscript and fixed typos and spelling errors.

Reviewer #2 (Remarks to the Author):

I had reviewed a previous version of the manuscript submitted to another journal. The authors had made extensive changes, and I think that the new manuscript, submitted to Nature Human Behaviour did a great job in presenting the results of this extremely important endeavor. The meta-analysis is extensive and informative. It provides a better understanding of the antecedents and consequences of goal engagement, disengagement and flexibility, and highlights promising avenues and research gaps. I especially appreciated how the idea of goal levels was introduced and addressed in the discussion (as a way to make sense of some of the findings). I expect this paper will thus be very valuable to the field. I had a few mostly smallish comments that I hope can help you further improve this manuscript.

We very much appreciate your overall positive evaluation of our work, and for your detailed and valuable feedback. In the lines that follow we have tried to address as many of your comments as we possibly could.

-it would be helpful to note early on (perhaps on p.4-5 when you describe different forms of goal adjustment) that disengagement and reengagement in this tradition (especially in the research by Wrosch & colleagues, where it is assessed as an individual difference) typically refers to general tendencies and is not about engagement with or disengagement from a specific goal. This has implications for contextualizing the existing research (so could be returned to in the discussion). There is a difference in what someone does for a specific goal

vs. the person's general tendencies; unfortunately the current research does not allow you to distinguish these (since most research focuses on tendencies), and this could be emphasized further in the introduction and discussion sections of the manuscript.

RESPONSE:

In agreement, we expanded accordingly both the Introduction and Discussion sections:

Introduction (pg. 5-6):

“Although some strands of the goal adjustment literature conceptualise disengagement and reengagement as trait-like tendencies (Wrosch, Carver, Scheier et al., 2003), other research has emphasised the situational and goal-specific nature of these processes (Ntoumanis et al., 2014; Riddell et al., 2022). This distinction between dispositional tendencies and context-dependent responses is critical for advancing theoretical and empirical understanding. Trait-level measures capture enduring individual differences in how people typically respond to goal-related challenges, offering valuable insights into broad patterns of self-regulation. However, such measures may overlook the dynamic, context-sensitive adjustments that individuals make in response to specific goal failures or life events, patterns more effectively captured by goal-specific assessments. A comprehensive understanding of goal adjustment therefore requires attention to both the predictors of stable tendencies and the situational factors that shape goal-specific behaviour.”

Discussion (pg. 31):

“These distinctions are consequential, as disengagement may hinder progress at one level, whereas reengagement can preserve or redirect progress toward higher-order goals. For example, experiment that manipulate goal attainability have demonstrated that individuals often make greater progress towards overarching aims when they relinquish unattainable lower-order goals (Choi et al., 2020; Ntoumanis et al., 2014; Riddell et al., 2022). Moreover, the boundaries between goal progress, disengagement, and reengagement are often more fluid than typically presented. An individual may, for instance, appear to abandon the goal of finding a romantic partner and redirect effort toward self-improvement, yet may not have fully disengaged from the original goal, even after months or years of nonpursuit. This conceptual ambiguity is reflected in emerging evidence on goal shelving and freezing phenomena (Davydenko et al., 2019; Mayer & Freund 2025). We contend that a more nuanced understanding of goal adjustment requires researchers to begin by precisely specifying the referent of disengagement and reengagement, namely, disengagement or reengagement with what?”

-I appreciate how you grouped the different variables into categories (rather than examine separate specific variables with few studies in each), and agree that this is the better approach. I would have liked, however, a separate table in manuscript that names every variable that went into each category (and how many studies it was in). Otherwise some of these categories seem pretty nebulous.

RESPONSE:

We provided separate tables naming the variable that went into each category (ST-4), as well as naming which studies examined which variables (ST-5), in Supplementary Information. We updated these tables to include added studies (following Reviewer's 1 request). We acknowledge that the reviewer's suggestion to include these tables in the manuscript would enhance their visibility. However, we note that these are very large tables and argue that the information contained therein is ancillary to the conceptual model presented in the manuscript. Thus, to streamline what is already a dense manuscript, we kept the tables in Supplementary Information. To help readers locate this material, we added relevant notes throughout the manuscript. Of course, if the reviewer or the editor feel strongly against this decision, we will reconsider.

-I think that too much attention is drawn in the text to potential moderators, especially since some of the results by moderator may instead be due to the specific variable examined. For example, if studies from Germany were more likely to include action crises (as opposed to another precursor variable), the moderation analysis would show stronger effects from Germany, where the strength of the effects is due to the specific variable examined. Having the information in the text also makes it a bit long, reducing the focus on the more important results. I suggest moving it to the tables instead (perhaps another column in tables 1 & 2 stating significant moderators of the effects).

RESPONSE:

We agree with the reviewer's conclusion that the moderators are over-emphasised and detract from the main results. As such, aligning with the reviewer's comment, we moved the moderator analyses to Supplementary Information (SI-1). We focused on the primary results in the manuscript.

-You present on p. 21 a nice discussion of your evaluation of publication bias. I wondered how problematic/impactful that bias may be? There are many different bias corrections that attempt to estimate effects corrected for bias, (some much more stringent/biased than others), so I am reluctant to recommend that you include some sort of corrected estimate in the table, but maybe a likely range of effects corrected for bias? Or explain why you're not doing it.

RESPONSE:

We concur that publication bias is a concern in meta-analytic research and appreciate the opportunity to elaborate on our approach.

Although methods such as trim-and-fill, selection models, and PET-PEESE offer ways to adjust for potential publication bias, they rely on strong and often unverifiable assumptions about the nature and mechanism of missing studies. These adjustments can be highly unstable, especially in the presence of heterogeneity, small numbers of studies, or non-standard patterns of bias, leading to misleading or overly speculative corrections. Moreover, the adjustments do not restore missing information but rather simulate it, which may compromise the transparency and replicability of the meta-analytic estimate. Therefore, it

may be more appropriate to interpret the unadjusted results cautiously and complement them with a narrative discussion of potential bias, rather than relying on potentially flawed statistical corrections.

For these reasons, we opted not to present bias-corrected estimates. Additionally, given the absence of consensus in the field regarding the most appropriate method, we were concerned that presenting a single or a range of corrected estimates might offer a false sense of precision or certainty.

Given these considerations, we transparently reported our evaluation of publication bias and acknowledged its potential influence, but refrained from presenting corrected effect size estimates (pg. 31).

“Although publication bias may have influenced the results of the current meta-analysis, we elected not to present bias-corrected estimates of effect sizes. Such corrections can convey a misleading sense of precision or certainty, given the lack of consensus on the most appropriate adjustment method and the sensitivity of different approaches to their underlying assumptions, which can yield markedly different results (Carter et al., 2019; McShane et al., 2016).”

Reviewer #3 (Remarks to the Author):

This manuscript presents a meta-analysis of both the antecedents and consequences of 3 forms of goal adjustment – goal disengagement, goal reengagement, and flexibility. It presents a novel conceptual organization of antecedent variables, which helps to create a tentative model of how different antecedent variables are related to different forms of goal adjustment. It also extends beyond prior meta-analyses (Cheng et al., 2014; Barlow et al., 2014) that focused on the implications of goal-adjustment for wellbeing outcomes, and also considers implications for Goal Attainment and “Functioning” – which appears to refer to physical health, social relationships, and work-related outcomes.

This manuscript got off to somewhat of a slow start, as I was initially unsure what it contributed beyond prior meta-analyses. But by the end of the introduction, the authors finally told me they were organizing research on the antecedents of goal-adjustment, and I was quickly pulled in and even enthralled by later portions of the paper. I spent far more time than I planned on this review, but actually enjoyed the experience greatly. Broadly, then, some version of this manuscript should be published, but there are some substantial issues that the authors will need to successfully attend to before this should be done.

We very much appreciate your overall positive evaluation of our work, and for your detailed and valuable feedback. In the lines that follow we have tried to address as many of your comments as we possibly could.

First, the authors need to overcome their “slow start”. For starters, the title and the abstract don’t tell the reader what people are “adjusting” their goals in response to (i.e., obstacles and

difficulties achieving one's goals), which would create a great deal of confusion about what the paper is even about for many potential readers. For me, this was an easy issue to overcome because I am familiar with this literature. But many potential readers won't be, and this will hamper the article's ability to impact the literature.

Second, neither the abstract nor the opening sections of the introduction clarify the unique contribution of the current meta-analysis beyond prior meta-analyses. This should be front and center from the very beginning. They organized potential antecedent variables, expanded the outcome variables under consideration to also include goal-progress and functioning, and used the results to create an integrative model. No prior meta-analysis has done that. Don't make me search until the end of the introduction to figure this out.

Relatedly, the authors should emphasize the relative dearth of theorizing on antecedents of goal-adjustment much earlier and more clearly. This is the main reason why this paper has the potential to contribute greatly to the literature.

RESPONSE:

We thank the reviewer for persisting with the manuscript (rather than disengaging) and are glad they ended up enjoying it. There is a lot of ground to cover in this review and engaging readers from the outset is, as the reviewer points out, critical for enhancing the manuscript's impact. The reviewer makes a valid point about the slow start and offers useful suggestions for addressing this issue. We revised the title and Abstract to clarify that we examine adjustment in response to difficulties during goal striving. Moreover, we rewrote the opening of the introduction (pg. 2):

“In the pursuit of personal and professional aspirations, individuals frequently encounter obstacles, setbacks, or shifting circumstances that challenge the attainability or desirability of their goals. In response to these difficulties, people may persist in their efforts, or they may adjust their goals—modifying, reprioritising, or abandoning them altogether. Although the benefits of persistence have long been emphasised (Locke & Latham, 2019), a growing body of research has highlighted the adaptive value of adjustment when goals become unattainable or overly costly to pursue (Brandstätter & Bernecker, 2022; Wrosch, Scheier, Carver, et al., 2003).

Despite this empirical growth, the literature on goal adjustment remains fragmented. Researchers have approached the topic from a variety of theoretical perspectives and have examined a wide range of potential predictors and outcomes of goal adjustment across diverse life domains. Yet, to date, meta-analytic efforts have primarily focused on associations between various goal adjustment capacities and wellbeing (Barlow et al., 2020; Cheng et al., 2014). This work, although important, has not considered other critical issues such as antecedents of goal adjustment, and effects of goal adjustment on goal striving and day-to-day physical or social functioning.

In this article, we address these knowledge gaps by presenting a meta-analytic synthesis and an integrative conceptual model of goal adjustment, its antecedents, and its outcomes. Specifically, we (i) organise and synthesise empirical findings on the antecedents and outcomes of goal adjustment across diverse domains; (ii) broaden the

scope of outcomes examined to include not only psychological wellbeing, but also goal striving, physical health, and social functioning; and (iii) use the resulting evidence base to propose a novel model of goal adjustment that integrates findings across multiple theoretical perspectives. In doing so, we provide a comprehensive synthesis that advances understanding of the factors that influence goal adjustment and its multifaceted outcomes.”

Beyond that, I would encourage the authors to break apart the “Functioning” and “Impairment” outcome variables more, to whatever extent is possible with the studies available for this meta-analysis. As I noted above, this appears to refer to a heterogeneous collection of variables, including physical health, social relationships, work-related outcomes, and possibly others (e.g., academic outcomes, I would venture to guess). These are of great interest to researchers in different fields – e.g., health psychology, relationship science, industrial/organization psychology; not to mention other fields entirely like behavioral health and management. While researchers in these fields might be truly enthralled to hear what this meta-analysis can tell them relevant to their outcome variables of interest, they might see very little value in the variable of “functioning”.

RESPONSE:

Functional outcomes are, as the reviewer points out, somewhat heterogeneous as they cover the gamut of attributes that enable individuals to live their day-to-day lives and that do not fall under “goal striving” or “mental wellbeing.” Nonetheless, we contend that these are important attributes to consider, both for their role in goal striving itself and because of proposed theoretical links between adaptive adjustment and changes to functional ability.

Functioning/impairment related outcomes have been studied less frequently than either goal striving or wellbeing (only ~6% of outcome-related effect sizes were related to functioning/impairment), making difficult to meaningfully divide this category further. Nonetheless, we agree with the reviewer that there is value in a more fine-grained approach to examining functioning. Consequently, we meta-analysed the individual variables comprising each outcome category (i.e., including functioning) and found (as well as reported) significant results.

Moreover, we extended our discussion on associations between functioning and goal adjustment at various places in the manuscript. We note all these changes as follows:

Introduction (pg. 6):

“Theoretical perspectives have also highlighted the relevance of goal adjustment for both physical and societal functioning. Across the lifespan, the ability to flexibly modify personal goals has been identified as a critical adaptive resource. For instance, goal adjustment plays a central role in coping with age-related declines in physical capacity (Brandtstädter & Renner, 1990; Heckhausen et al., 2010), the onset of chronic health conditions (Cho et al., 2023), and in the successful management of social relationships in adulthood (Greve et al., 2014). Similarly, adaptive goal adjustment has been linked to effective preparation for retirement (Liu et al., 2024) and to navigating career transformations (Tolentino et al., 2013). The capacity to recalibrate or disengage from obstructed goals during major life transitions or

unexpected events enables individuals to maintain functional autonomy and psychological well-being in a rapidly changing world. Conversely, failure to adapt one's goals in the face of shifting circumstances can compromise one's ability to manage emergent demands across the life course."

In Discussion (pg. 27):

"Our meta-analytic results revealed that goal striving flexibility enhances both wellbeing and functioning, while protecting against illbeing, highlighting its multifaceted role in promoting health and wellbeing. The benefits of maintaining a flexible disposition toward goal adjustment appear to be widespread, with positive implications for mental health, physical health, and social functioning."

More modest issues with introduction:

Haase et al. (2013) reviewed and integrated an overlapping set of theories and arrived at 3 very similar constructs to Scobbie et al. (2021) (which the authors use to organize research on different forms of goal-adjustment). This should be cited and acknowledged. Currently, this seems to be cited as an extension of Brandtstadter's model, but it is an integration with other models.

RESPONSE:

We agree that including a discussion of Haase and colleague's work would enhance the relevant section. We did so as follows (pg. 3):

"An alternative approach taken by Haase et al. (2013) aggregated the dual process model (Brandtstädter & Rothermund, 1990), the model of selection, optimisation, and compensation (Freund & Baltes, 2002), and the motivational theory of lifespan development (Heckhausen et al., 2010) to highlight three comparable constructs: goal engagement, goal disengagement, and metaregulation (flexibly optimising goals to match changing circumstances and opportunities). Their work underscores the dynamic interplay between persistence and flexibility, and reflects a broader attempt to unify perspectives on how individuals regulate their goals over the life course. Although Scobbie et al. (2021) offered a structured taxonomy for organising research on adjustment, Haase et al. (2013) provided a theoretically integrated account of core self-regulatory mechanisms.

Building on these research efforts, we propose that three core adjustment processes capture how individuals altering their goal and their general propensity to adjust goals. We use *goal adjustment* to refer to the overarching concept that encompasses the three processes: *goal disengagement*, *goal reengagement*, and *goal striving flexibility*. We adhere to the definitions of goal disengagement and reengagement presented in the previous paragraphs, and define goal striving flexibility as an individual's general disposition to remain open to modifying their striving, cognitions, or goals to better align with their capabilities, resources, and situational demands. These processes are broadly captured by the various theoretical perspectives that have been put forth to explain goal adjustment."

Page 5-6: The authors say that changes in goal-striving are the mechanism that links goal adjustment to wellbeing and cite Carver & Schier (2005), but I don't follow. Disengagement and reengagement are both seen as beneficial for wellbeing, but they should (theoretically speaking) affect goal-striving in opposite directions, correct? The example previously prior to this sentence seem to involve interactions between goal-disengagement/reengagement and goal-achievement/failure in predicting wellbeing, which the authors don't consider in this meta-analysis.

RESPONSE:

We can see that this sentence is confusing. We meant to convey that disengaging and reengaging are the means by which individuals improve their wellbeing (i.e., persisting with a failing goal will not result to wellbeing). However, as the reviewer pointed out, testing this statement would likely require testing meta-analytic interactions or mediation analyses, which is beyond the scope of this manuscript. We therefore removed the statement.

Methods:

I'm not a methodological expert in systematic review or meta-analysis, but have of course read enough of them over my career. The authors' procedures seemed largely appropriate, but I encourage the editor to gather reviews from someone with greater methodological expertise here. I would defer to their opinion on these issues.

Question of Causal Direction: Affect is considered as both a possible antecedent of goal-adjustment (Affect-Enhancing, Affective-Diminishing) and as a possible outcome of goal-adjustment (Wellbeing, Illbeing). The same is also true of Goal Progress – i.e., Goal-Enhancing/Inhibiting antecedents (and possibly Agency?) vs. Goal-Attainment/Failure outcomes. When a study included in this meta-analysis examined these variables experimentally or longitudinally, it is quite straightforward to code specific effects for inclusion into one analysis or the other. But the current authors also note that they included cross-sectional correlational studies, and that they seem to have used the original papers' authors hypotheses to sort effects into the antecedent vs. outcome meta-analyses. First, is that an accurate description of the current procedures? If so, this seems potentially problematic, as the authors of the original articles could of course be wrong and the effects may pertain to the opposite direction of causality in reality. This could spuriously increase effects that should only be listed in one direction of causality (or perhaps neither), but are currently listed in both directions. Looking at Figure 2, Affect-Enhancing antecedents are related to both Reengagement and Flexibility, and, likewise, Wellbeing consequences are related to both Reengagement and Flexibility. Likewise, Goal-Enhancing antecedents, Agency-Enhancing antecedents, and Goal-Attainment outcomes are all related to Goal Reengagement. I'd like to see the authors conduct/report analyses (in the paper itself, not in the supplement) that examine if longitudinal and/or experimental studies provide clear evidence for a bidirectional effect or if these effects are actually unidirectional (or even consistently non-significant) after removing cross-sectional correlational studies from the analysis.

RESPONSE:

To address the reviewer's first question: Yes, we included cross-sectional correlational studies and used the primary study's hypotheses to sort effects into antecedents and outcomes. We recognise that this, in some way, is not ideal as it presumes the primary study authors are correct in their assertion that the variables are antecedents. We dealt with this issue in the Discussion section of the original submission. We think that the reviewer's suggestion to report an analysis of longitudinal and experimental studies is an excellent solution for examining true antecedent and outcome effects, and have done so (see Results section). These analyses largely support the proposed temporal structure of our model (i.e., antecedent variables proceed goal adjustment and goal adjustment predicts outcomes). Even though our sensitivity analysis provides some initial evidence for directional associations in goal adjustment, we noted that this evidence does not preclude the possibility of bidirectionality. Indeed, some of our conclusions and others' theorising (Heckhausen et al., 2010) would suggest that successfully adjusting goals produces improvements in outcomes, which in turn set the stage for the development of conditions/dispositions that contribute to adjustment.

To address this issue, longitudinal research is needed using implementing techniques that account for interdependencies across time (e.g., dynamic structural equation modelling; McNeish & Hamaker, 2020). See our addition to the Discussion (pg. 25):

“Addressing the directionality and complexity of associations among antecedents, goal adjustment capacities, and outcomes may be best accomplished through lifetime longitudinal or cohort studies, coupled with advanced statistical techniques that account for interdependencies over time, such as dynamic structural equation modelling (McNeish & Hamaker, 2020).

Results and Discussion:

The organization of different antecedent variables in the “Conceptual Model” section is quite useful, as is Figure 2!

Tables: Why are the variables in Table 1 (and Table 2) listed in different orders for different goal-adjustment variables? Significant effects appear to be aggregated at the top of each panel (except for Affect Enhancing-Flexibility relationship in Table 1, oddly), but they're not in order of effect size or alphabetical otherwise. I suggest a more systematic approach, so readers looking for a particular variable can find it more easily. Perhaps place the antecedent/outcome variables in alphabetical order, and highlight significant effects so that they “pop out” for readers who are just interested in finding significant effects (e.g., star the significant effects or put them in bold font).

It would be useful if the Results section text told the reader in which section of the Supplementary Information they can find specific moderator analyses – e.g., the Antecedents of Goal Disengagement results section could tell me where to find the effects for other goal-inhibiting variables besides Action Crisis (if there were any available), and the methodological moderation analysis (comparing lab, longitudinal, experience-sampling studies). I was unable to find or locate the more detailed presentation of the moderation analyses.

RESPONSE:

In line with the reviewer's comment, we reorganised the tables so that effects are in order of effect size and significant effects are highlighted. We also added pointers throughout the Results section to assist readers in finding information in the supplementary tables. We moved results pertaining of the moderation analyses to Supplementary Information (SI-1), aligning with the recommendation of Reviewer 2. We had provided summary p-values for moderators in supplementary tables (ST-1) and a script on the project's OSF page that enabled users to run the analysis and obtain the full output. In addition, we tabulated output for the moderator analyses in Supplementary Information (SI-1).

P16: Antecedents of Goal Reengagement section: Optimism is listed as an affect-enhancing variable here, but the "Conceptual Model" section above lists it as an adaptive personality trait. Is there perhaps a distinction being made here between Dispositional/Trait Optimism and Goal-Specific Optimism? If so, more precise terminology should be adopted to clarify this. Or is this just a mistake?

RESPONSE:

We coded "optimism" as an affect-enhancing variable. We made the correction (pg. 17).

It is odd that "Goal-Inhibiting" antecedent variables are associated with greater Goal Reengagement. More explanation is required, in terms of the variables that constitute this antecedent category. This is somewhat complicated by the fact that no individual Goal-Inhibiting variable alone is linked with Goal-Reengagement, but more information is needed nonetheless. For example, I could imagine that an Action Crisis with Goal #1 could lead to Goal-Reengagement with Goal #2 (presumably after one also disengaged from Goal #1). Do all (or even most) of the Goal-Inhibiting variables refer to the previous goal (i.e., Goal #1) and not to the Reengaged goal (i.e., Goal #2)?

RESPONSE:

We also pondered this issue. It largely stems from researchers often measuring "reengagement" but failing to articulate "with what." As a consequence, we cannot provide a confident answer to the reviewer's question based on the primary literature. We regard this as a limitation of the field.

As the reviewer pointed out, action crisis is a good example of a goal inhibiting variable (i.e., something associated with declining effort and performance on the original goal) that might be positively related to reengagement if the individual were to find a new goal that serves the same higher order purpose as the original. Action crisis helps the individual let go of their old mode of striving and find something new that fulfils the same need. Essentially, goal inhibiting antecedents make room for reengagement to occur. We amended the Discussion section to better articulate how both goal enhancing and inhibiting antecedents might contribute to goal reengagement, using examples recommended by the reviewer (pg. 27):

"To illustrate this point, consider the impact an action crisis on goal striving processes. On the one hand, the experience of an action crisis may lead to reduced commitment and effort toward the original goal (Brandstätter et al., 2013), decreasing the likelihood that the individual will reengage with that same goal in the future,

potentially due to processes such as goal devaluation (Heckhausen & Schulz, 1995). In this context, a researcher assessing reengagement with the original goal may observe a negative association between reengagement and indicators of commitment or effort. On the other hand, an action crisis can also prompt disengagement from the unattainable goal and facilitate a shift in focus toward a new, more attainable goal (Vann et al., 2018). Consequently, a researcher examining reengagement with alternative goals would likely find a positive association between reengagement and commitment or effort. These divergent patterns highlight the importance of specifying the target of reengagement (i.e., original vs. new goals) when interpreting findings on goal adjustment processes.”

The authors attempt to address this issue in the Discussion section (on page 25 of the pdf). I honestly had some difficulty following this discussion section, so, first, I encourage the authors to use examples to make their discussion clearer. For example, they state: “Goal reengagement encompasses both the continuation of goal striving at a high level of one’s goal hierarchy and the discontinuation and subsequent adjustment of striving at lower levels.” To flesh this out, perhaps a concrete example would be useful – e.g., perhaps a person still wishes to exercise and improve their physical health. An injury prevents them from running anymore, so they switch to swimming? (Not entirely sure if that corresponds to their intended, abstract description.) The next sentence reads, “Viewed in this light, it follows that both processes involved in the cessation of old modes of goal striving and processes that reinforce striving for the goal at a higher conceptual level contribute to goal reengagement.” If I’m following the authors correctly, does this mean that they hypothesize that “Goal-Enhancing” antecedent variables related to the Higher-Order Exercising goal (in the previous example) would be related to Reengagement (but not the “Goal-Enhancing” variables related to the Lower-Order Running goal). Likewise, does it mean that they hypothesize “Goal-Inhibiting” variables related to the Lower-order Running goal are related to Reengagement (but not “Goal-Inhibiting” variables related to the Higher-Order Exercise goal)? If this is the case, I again wonder if the authors can examine the actual studies in this meta-analysis to determine if this hypothesis is supported. A formal moderation meta-analysis would be preferable if possible, of course, but even more informal observations would be useful here. I also wonder if the authors will need to consider whether Goal-Inhibiting variables are related to two different Lower-Order goals (e.g., running vs. swimming in my prior example; or the previous and the new lower-order goal more broadly), and if these two lower-order goals are always necessarily related to the same higher-order goal at all?

RESPONSE:

We are hesitant to conduct post-hoc moderation analyses, as they offer limited explanatory power and spurious results (Cochrane Handbook, Chapter 10; <https://training.cochrane.org/handbook/current/chapter-10#section-10-10>). Additionally, as noted above, it is not always clear what goals authors examine when they measure reengagement in primary studies. This makes it difficult to answer the question that the reviewer poses as to whether goal enhancing/inhibiting antecedents relate to higher or lower order goals. Observationally, a few studies did experimentally examine goal progress after participants abandoned a failing goal and reengaged with a higher order goal that served the

same purpose, and tended to find that participants made more progress, particularly if the unattainability of the goal was recognised early (Choi et al., 2020; Ntoumanis et al., 2014; Riddell et al., 2022). We agree that further consideration of this topic is necessary and have amended the Discussion section to include observations from individual studies. Further, we called for researchers to start addressing the questions “disengagement from what” and “reengagement with what” (pg. 30-31):

“A key limitation in the current literature is the lack of clarity regarding the level of the goal hierarchy at which disengagement and reengagement are assessed. Adjustment processes may differ markedly depending on whether they occur within the same hierarchical level (e.g., shifting from regular running to walking) or across levels (e.g., from exercising to dieting to achieve weight loss). These distinctions are consequential, as disengagement may hinder progress at one level, whereas reengagement can preserve or redirect progress toward higher-order goals. For example, experiments that manipulate goal attainability have demonstrated that individuals often make greater progress towards overarching aims when they relinquish unattainable lower-order goals (Choi et al., 2020; Ntoumanis et al., 2014; Riddell et al., 2022). Moreover, the boundaries between goal progress, disengagement, and reengagement are often more fluid than typically presented. An individual may, for instance, appear to abandon the goal of finding a romantic partner and redirect effort toward self-improvement, yet may not have fully disengaged from the original goal, even after months or years of nonpursuit. This conceptual ambiguity is reflected in emerging evidence on goal shelving and freezing phenomena (Davydenko et al., 2019; Mayer & Freund 2025). We contend that a more nuanced understanding of goal adjustment requires researchers to begin by precisely specifying the referent of disengagement and reengagement, namely, disengagement or reengagement with what?”

Regarding the surprising null relationship between Goal-Reengagement and Goal Progress Outcomes: On page 25-26 of the discussion, the authors note (similar to the previous point) that this could be due different studies examining goal progress for different goals (i.e., the original vs. reengaged goal) at different times (before vs. after reengagement). I again wonder if the authors can code goal-progress outcomes to shed some light on this issue. Perhaps some of the goal-progress measures are too non-specific to be coded in this manner, but if there are a sufficient number that can be coded in this manner, it would be extremely useful.

RESPONSE:

We agree with the reviewer that this analysis would be useful. Unfortunately, as articulated in previous responses, we found that largely goal progress measures were too non-specific to examine the effects of goal reengagement in more depth.

The effects of Goal-Disengagement on “Impairment” need to be broken apart in the main results section more. As I mentioned in relation to the introduction and the framing of the paper more broadly, Health psychologists, Relationship scientists, and I/O psychologists will want to know if these effects pertain to their domain of interest.

Moreover, this effect is conceptually important to break apart and understand further. The effect of Goal Disengagement on Goal-Progress is basically tautological by itself. But understanding the effects of Disengagement on Impairment goes beyond mere goal progress and could highlight ways in which Disengagement truly is a very mixed bag. Sure, it might lower Depression and Anxiety (Illbeing), but these results seem to imply it MIGHT simultaneously reduce one's lifespan, result in getting fired from one's job, and/or lead to a divorce, perhaps???? If this is the case, Goal-Disengagement appears more similar to some societal stereotypes of the indifferent slacker – i.e., who is unemployed, single, content to live in his parents' basement till age 40, eating too many Cheetos, playing video games nonstop, etc.; rather than the more positive portrait that Wrosch and others paint. I was especially surprised to see that, later in the article, the authors did not discuss the relationship between Disengagement and Impairment in the relevant discussion section.

RESPONSE:

As we did for antecedent variables, we broke down outcomes into individual variables and added statistically significant relations to the Results section. Although we found an association between disengagement and the impairment category in our main analysis, we did not find any individual impairment variables associated with disengagement, nor did we find evidence of this association when examining only longitudinal and experimental studies. We offer two suggestions as to why this may be the case. First, as we noted in responses to other reviewers, impairment and functioning were the most sparsely investigated outcomes. Put simply, reducing the pool of effects examining only longitudinal/experimental studies or a single individual variable left us with little to synthesise. Second, as highlighted by the reviewer in a previous comment, there is the possibility of bidirectionality in cross-sectional studies. It is likely that an individual disengages when they experience impairment. In the case of, say, chronic illness, this impairment might remain, but disengaging from unattainable goals associated with the illness may set the stage for a more fulfilling life in the long run. Of course, these are speculations, and we can offer little evidence for either case. We proceeded to mention the finding in the Discussion but pointed out that the result should be interpreted with caution (pg. 29).

“We found little evidence linking goal disengagement or reengagement to better functioning. One explanation is that improvements in functioning require time to manifest and may only become apparent after individuals achieve success with newly adopted goals. This interpretation aligns with our finding that dispositional flexibility, rather than more proximal disengagement or reengagement, more strongly predicts functioning. Notably, we observed a positive association between disengagement and impairment. Although this could reflect a “dark side” of disengagement—where letting go of goals offers short-term relief but risks longer-term purposelessness and dysfunction (Carver & Scheier, 2005)—this pattern was not evident in longitudinal or experimental studies. An alternative explanation is that the association is bidirectional, with impairment potentially prompting disengagement as a reactive strategy. Given these complexities, we advice caution in interpreting this finding and highlight the need for further research to disentangle the temporal and causal dynamic underlying the relationship between disengagement and impairment.”

More broadly, the individual outcome variables constituting each outcome category should be consistently broken apart (whenever possible), as the authors did for the antecedent variable categories. This could also shine further light on the Flexibility-Functioning relationships, for example.

RESPONSE:

In line with the reviewer's suggestion, we broke down the individual outcome variables in each category (see Results section) as we did for the antecedent variables. As mentioned in our response above, evidence for the disengagement-impairment association is mixed, and we thus tempered our pertinent discussion.

Finally, the Supplemental Table (with file name ending in 262818) listing the specific antecedent variables in each category was useful, but it appears to be missing Affect Enhancing and Affect Diminishing variables. I also happened to notice that "Type D Personality" is listed as an Adaptive Personality variable, but also an Agency Enhancing variable. (Which I only noticed because I have never heard of Type D personality before. I've heard of Type A and Type B, but not D or even C.) Also, Optimism is not listed as an Adaptive Personality variable, but is in the text.

Regardless, the important points are – are the authors actually counting variables in two or more categories? This seems inappropriate, as it would muddy the literature rather than clarify it. Or are there just simply mistakes in various places that need to be corrected (hopefully only in the paper itself, but if there are mistakes in coding of the meta-analysis itself, some aspects of it would of course need to be re-done)?

RESPONSE:

Another reviewer raised the same issue. We responded as follows:

The Supplementary Information table containing the categorisations belonged to an earlier (i.e., pre-submission) draft and, as the reviewer noted, there were some inconsistencies. Thankfully, these inconsistencies were not mirrored in the main data file; that is, the analysis was based on our finalised categorisation scheme. We corrected the table (see ST-4).

Response to Reviewers

Reviewer #1 (Remarks to the Author):

Thank you for your careful consideration of my comments. You have addressed these in the review

Reply: We thank the reviewer for taking the time to review our manuscript and for their helpful input in the previous round.

Reviewer #2 (Remarks to the Author):

The authors did an excellent job addressing all my comments. I have one more small lingering observation: In the definitions of the key constructs (engagement, disengagement, flexibility), both engagement and disengagement are defined as processes (p. 3, p. 5), while flexibility is defined as “an individuals’ general disposition” (p. 3 & 4) – it would be helpful if the authors explicitly acknowledged this discrepancy and discussed potential implications.

Reply: We thank the reviewer for their positive evaluation of our revision and for their helpful suggestion. We have adjusted the introduction to acknowledge this distinction and highlight the implications it has had for the study of goal adjustment.

This distinction has implications for how these constructs have been studied in the literature. Engagement, disengagement, and reengagement are often assessed with respect to specific goals at the behavioural or situational level. That is, authors are often concerned with how individuals react to unattainability in a certain domain or situation. Conversely, examinations of flexibility have been more interested in how people adapt to setbacks generally or over a protracted timescale. Disengagement, reengagement, and flexibility have been broadly captured by various theoretical perspectives that have been put forth to explain goal adjustment. However, by focusing on disengagement, reengagement, and flexibility as discreet processes, these accounts have somewhat overlooked the potential interplay between dispositional characteristics and situational goal adjustment. An integrated perspective is needed to clarify the similarities and differences among these goal adjustment processes.

Reviewer #3 (Remarks to the Author):

I was previously listed as Reviewer #3. This revision is very responsive to many of my previous concerns, so I thank the authors for their efforts and diligence in this! The clarification of the purpose of the meta-analysis and conceptual model earlier on is helpful, as is the addition of sensitivity analyses focused on experimental/longitudinal studies to better establish the order of causality. I appreciated the authors' point that many of the questions I posed to them are ultimately issues in the broader literature – authors need to do a better job of specifying whether disengagement/reengagement/goal-progress/etc. is linked to an original goal or an alternative goal. This should help to improve research practices in this area moving forward (hopefully, of course).

Reply: we thank the reviewer for their positive feedback on our revision.

My only remaining feedback is that I thought the authors could “go further” in addressing some of my original concerns. In most cases, this is minor, so I will leave it to the editor's discretion (of course) whether to require this or not. In all cases, the goal is clear presentation to help the reader better understand the paper, rather than a substantive concern with the underlying science.

-The opening paragraphs now do an excellent job of highlighting the novel purpose of this paper. The revised title and abstract are certainly improved, but they are still missing some of the critical emphasis that comes across in the opening paragraphs. I realize it's difficult to fit everything in a short title and short abstract, but I would encourage the authors to further revise these to better highlight the novel contribution – i.e., the fact that they are presenting a novel conceptual model of antecedent variables and extending the outcomes considered beyond wellbeing to also include goal-progress and functioning.

Reply: We agree that we can further emphasise the contribution of the paper in the abstract and title and have made amendments to highlight the contributions of our work. We do, however, note that we must remain within the word limits for the journal and adhere to the reporting policy, which precludes authors from making novelty claims about their work (<https://www.nature.com/articles/s41562-021-01068-x>).

-In their response letter, the authors indicate that “We reorganised the tables so that effects are in order of effect size and significant effects are highlighted”. This is certainly true for Table 1, but it does not appear to be completely true for Table 2. That is, significant effects are highlighted in Table 2, but outcome are now listed in a standardized order for Table 2. I encourage the authors to use a standard system across tables to help the reader find variables of interest to them.

Reply: We have amended Table 2 to follow the same organisation as Table 1.

-Finally, I vaguely recall that it was easier to figure out what common measures of goal engagement, disengagement, and flexible striving were for the included studies, in the prior version of the paper (so I did not note this as an issue). But I had more trouble finding out what they were in this version. Are most studies (that used a standardized, trait-like measure of these tendencies) based on Wrosch et al. (2003) and Brandtstadter & Renner (1990)’s scales, for example?

Reply: This information has now been added to the manuscript

Effects related to goal disengagement (64%) and reengagement (77%) were predominantly measured with the respective subscales of the Goal Adjustment Scale (Wrosch, Scheier, Carver, et al., 2003), while effects related to goal striving flexibility (78%) were predominantly measured using the flexible goal adjustment subscale of the Brandtstädter and Renner’s (1990) Tenacious Goal Pursuit/Flexible Goal Adjustment scale.